# Iteration and Stochastic First-order Oracle Complexities of Stochastic Gradient Descent using Constant and Decaying Learning Rates

## Abstract

The performance of stochastic gradient descent (SGD), which is the simplest first-order optimizer for training deep neural networks, depends on not only the learning rate but also the batch size. They both affect the number of iterations and the stochastic first-order oracle (SFO) complexity needed for training. In particular, the previous numerical results indicated that, for SGD using a constant learning rate, the number of iterations needed for training decreases when the batch size increases, and the SFO complexity needed for training is minimized at a critical batch size and increases once the batch size exceeds that size. This paper studies the relationship between batch size and the iteration and the SFO complexities needed for nonconvex optimization in deep learning with SGD using constant/decay learning rates. We show that SGD using a step-decay learning rate and a small batch size reduces the SFO complexity to find a local minimizer of a loss function. We also provide numerical comparisons of SGD with the existing first-order optimizers and show the usefulness of SGD using a step-decay learning rate and a small batch size.

## 1 Introduction

### 1.1 Background

First-order optimizers can train deep neural networks by minimizing loss functions called the expected and empirical risk. They use stochastic first-order derivatives (stochastic gradients), which are estimated from the full gradient of the loss function. The simplest first-order optimizer is stochastic gradient descent (SGD) (Robbins & Monro, 1951; Zinkevich, 2003; Nemirovski et al., 2009; Ghadimi & Lan, 2012; 2013) and it has a number of variants, such as momentum methods (Polyak, 1964; Nesterov, 1983) and adaptive methods including adaptive gradient (AdaGrad) (Duchi et al., 2011), root mean square propagation (RMSProp) (Tieleman & Hinton, 2012), adaptive moment estimation (Adam) (Kingma & Ba, 2015), adaptive mean square gradient (AMSGrad) (Reddi et al., 2018), and Adam with decoupled weight decay (AdamW) (Loshchilov & Hutter, 2019).

SGD can be applied to nonconvex optimization (Vaswani et al., 2019; Fehrman et al., 2020; Chen et al., 2020; Scaman & Malherbe, 2020; Loizou et al., 2021; Arjevani et al., 2023; Khaled & Richtárik, 2023), where its performance strongly depends on the learning rate $\alpha_k$. For example, under the bounded variance assumption, SGD using a constant learning rate $\alpha_k = \alpha$ satisfies that $\frac{1}{K} \sum_{k=0}^{K-1} \|\nabla f(\boldsymbol{\theta}_k)\|^2 = O(\frac{1}{K}) + \sigma^2$ (Scaman & Malherbe, 2020, Theorem 12) and SGD using a decaying learning rate (i.e., $\alpha_k \to 0$) satisfies that $\frac{1}{K} \sum_{k=0}^{K-1} \mathbb{E}[\|\nabla f(\boldsymbol{\theta}_k)\|^2] = O(\frac{1}{\sqrt{K}})$ (Scaman & Malherbe, 2020, Theorem 11), where $(\boldsymbol{\theta}_k)_{k \in \mathbb{N}}$ is the sequence generated by SGD to find a local minimizer of $f$, $K$ is the number of iterations, and $\sigma^2$ is the upper bound of the variance.

The performance of SGD also depends on the batch size $b$. Convergence analyses of SGD in (Jain et al., 2018; Cotter et al., 2011; Chen et al., 2020; Arjevani et al., 2023) indicated that SGD with a decaying learning rate and large batch size converges to a local minimizer of the loss function. In (Smith et al., 2018), it was numerically shown that using an enormous batch leads to reductions in the number of parameter updates and model training time.

## 1.2 MOTIVATION

The previous numerical results in (Shallue et al., 2019) indicated that, for SGD using constant/linear decay learning rates, the number of iterations $K$ needed to train a deep neural network decreases when the batch size $b$ increases. Motivated by the numerical results in (Shallue et al., 2019), we decided to clarify *the iteration complexity* of SGD using a constant/decay learning rate needed to train a deep neural network in theory. The theoretical performance measure of SGD is $\min_{k \in [0:K-1]} \mathbb{E}[\|\nabla f(\boldsymbol{\theta}_k)\|] \leq \epsilon$, where $\epsilon \, (> 0)$ is the precision and $[0:K-1] := \{0, 1, \ldots, K-1\}$, which was used in the previous theoretical analyses of SGD. If SGD is an $\epsilon$–approximation $\min_{k \in [0:K-1]} \mathbb{E}[\|\nabla f(\boldsymbol{\theta}_k)\|] \leq \epsilon$, then SGD can train a deep neural network in $K$ iterations.

In addition, the numerical results in (Shallue et al., 2019) indicated an interesting fact wherein diminishing returns exist beyond a critical batch size; i.e., the number of iterations needed to train a deep neural network does not strictly decrease beyond the critical batch size. Here, we define *the stochastic first-order oracle (SFO) complexity* as $N := Kb$, where $K$ is the number of iterations needed to train a deep neural network and $b$ is the batch size, as stated above. The deep neural network model uses $b$ gradients of the loss functions per iteration. The model has a stochastic gradient computation cost of $N = Kb$. From the numerical results in (Shallue et al., 2019, Figures 4 and 5), we can conclude that using the critical batch size $b^\star$ (if it exists) is useful for SGD, since the SFO complexity $N(b)$ is minimized at $b = b^\star$ and the SFO complexity increases once the batch size exceeds $b^\star$. Hence, on the basis of the first motivation stated above, we decided to clarify the SFO complexities of SGD using constant/decay learning rates needed to achieve an $\epsilon$–approximation.

## 1.3 CONTRIBUTION

### 1.3.1 UPPER BOUND OF THEORETICAL PERFORMANCE MEASURE

To clarify the iteration and SFO complexities of SGD needed to achieve an $\epsilon$–approximation, we first give upper bounds of $\min_{k \in [0:K-1]} \mathbb{E}[\|\nabla f(\boldsymbol{\theta}_k)\|^2]$ for SGD generating the sequence $(\boldsymbol{\theta}_k)_{k \in \mathbb{N}}$ using constant/decay learning rates, as indicated in Table 1 (see Theorem 3.1 for the definitions of $C_i$ and $D_i$). The aim of this paper is to show that SGD is an $\epsilon$–approximation $\min_{k \in [0:K-1]} \mathbb{E}[\|\nabla f(\boldsymbol{\theta}_k)\|^2] \leq \epsilon^2$. Hence, it is desirable that the upper bounds of $\min_{k \in [0:K-1]} \mathbb{E}[\|\nabla f(\boldsymbol{\theta}_k)\|^2]$ become small. Table 1 indicates that the upper bounds become small when the number of iterations and batch size are large. In particular, it shows that a step-decay learning rate (the "Step Decay" row) may perform better other learning rates in the sense of minimizing the upper bound of $\min_{k \in [0:K-1]} \mathbb{E}[\|\nabla f(\boldsymbol{\theta}_k)\|^2]$. For example, if we set small batch size, such as $b = 2^1, 2^2$, SGD using a step-decay learning rate has the convergence rate, $\min_{k \in [0:K-1]} \mathbb{E}[\|\nabla f(\boldsymbol{\theta}_k)\|^2] = O(\frac{1}{K})$, which is better than the convergence rate $O(\frac{1}{K} + C_2) = O(\frac{1}{K} + \sigma^2)$ of SGD using a constant learning rate, where $\sigma^2$ is the upper bound of the variance. The table also indicates that the convergence of SGD strongly depends on the batch size, since the variance terms (including $\sigma^2$ and $b$; see Theorem 3.1 for the definitions of $C_2$, $D_2$, and $D_3$) in the upper bounds of $\min_{k \in [0:K-1]} \mathbb{E}[\|\nabla f(\boldsymbol{\theta}_k)\|^2]$ decrease as the batch size becomes larger.

### 1.3.2 OPTIMAL BATCH SIZE TO REDUCE SFO COMPLEXITY

Section 1.3.1 showed that using large batch sizes is appropriate for SGD in the sense of minimizing the upper bound of the performance measure. We are interested in finding appropriate batch sizes from the viewpoint of the computation cost of SGD. This is because the SFO complexity increases when batch sizes are sufficiently large. As indicated in Section 1.2, the critical batch size $b^\star$ minimizes the SFO complexity, $N = Kb$. Hence, we will investigate the properties of the SFO complexity $N = Kb$ needed to achieve an $\epsilon$–approximation. For example, let us consider SGD using a constant learning rate. Then, from the "Upper Bound" row in Table 1, we have that

$$\min_{k \in [0:K-1]} \mathbb{E}[\|\nabla f(\boldsymbol{\theta}_k)\|^2] \leq \underbrace{\frac{C_1}{K} + \frac{C_2}{b} \leq \epsilon^2}_{\Leftrightarrow K \geq K(b) := \frac{C_1 b}{\epsilon^2 b - C_2} \left( b > \frac{C_2}{\epsilon^2} \right)}.$$

We can check that the number of iterations, $K(b) := \frac{C_1 b}{\epsilon^2 b - C_2}$, needed to achieve an $\epsilon$–approximation is monotone decreasing and convex with respect to the batch size $b$ (Theorem 3.2). Then, we have

Table 1: Upper bounds of $\min_{k\in[0:K-1]} \mathbb{E}[\|\nabla f(\boldsymbol{\theta}_k)\|^2]$ for SGD using constant/decay learning rates and optimal batch sizes to minimize the SFO complexities ($C_i$ and $D_i$ are positive constants, $K$ is the number of iterations, $b$ is the batch size, and $L$ is the Lipschitz constant of $\nabla f$)

| Learning Rate | | Upper Bound | Optimal Batch Size |
|---|---|---|---|
| Constant $\alpha \in (0, \frac{2}{L})$ | | $\dfrac{C_1}{K} + \dfrac{C_2}{b}$ | $\dfrac{2C_2}{\epsilon^2}$ |
| | $a \in (0, \frac{1}{2})$ | $\dfrac{D_1}{K^a} + \dfrac{D_2}{(1-2a)K^a b}$ | $\dfrac{(1-a)D_2}{a(1-2a)D_1}$ |
| Decay | $a = \frac{1}{2}$ | $\dfrac{D_1}{\sqrt{K}} + \left(\dfrac{1}{\sqrt{K}} + 1\right)\dfrac{D_2}{b}$ | Small Batch Size |
| $\alpha_k = \frac{1}{(k+1)^a}$   $a \in (\frac{1}{2}, 1)$ | | $\dfrac{D_1}{K^{1-a}} + \dfrac{2aD_2}{(2a-1)K^{1-a}b}$ | $\dfrac{2a^2 D_2}{(1-a)(2a-1)D_1}$ |
| Step Decay $\alpha_k \geq \underline{\alpha}$ | | $\dfrac{D_1}{\underline{\alpha}K} + \dfrac{D_3}{\underline{\alpha}Kb}$ | Small Batch Size |

that $K(b) \geq \inf\{K\colon \min_{k\in[0:K-1]} \mathbb{E}[\|\nabla f(\boldsymbol{\theta}_k)\|] \leq \epsilon\}$, where SGD using the batch size $b$ generates $(\boldsymbol{\theta}_k)_{k=0}^{K-1}$. Moreover, we find that the SFO complexity is $N(b) = K(b)b = \frac{C_1 b^2}{\epsilon^2 b - C_2}$. The convexity of $N(b) = \frac{C_1 b^2}{\epsilon^2 b - C_2}$ (Theorem 3.3) ensures that a critical batch size $b^\star = \frac{2C_2}{\epsilon^2}$ whereby $N'(b^\star) = 0$ exists such that $N(b)$ is minimized at $b^\star$ (see the "Optimal Batch Size" row in Table 1). A similar discussion guarantees the existence of a critical batch size for SGD using a decaying learning rate $\alpha_k = \frac{1}{(k+1)^a}$, where $a \in (0, \frac{1}{2})$ or $a \in (\frac{1}{2}, 1)$ (see the "Optimal Batch Size" row in Table 1).

Meanwhile, for a decaying learning rate $\alpha_k = \frac{1}{\sqrt{k+1}}$ or a step-decay learning rate, although $N(b)$ is convex with respect to $b$, we have that $N'(b) > 0$ for all $b > 0$ (Theorem 3.3(iii)). Hence, for these two cases, a critical batch size $b^\star$ defined by $N'(b^\star) = 0$ does not exist. Accordingly, *small* batch sizes are appropriate for a decaying learning rate $\alpha_k = \frac{1}{\sqrt{k+1}}$ or a step-decay learning rate in the sense of minimizing the SFO complexities. Accordingly, we will define the *optimal batch size* (in the sense of minimizing the SFO complexity) by

$$\text{Optimal Batch Size } b^* = \begin{cases} \text{Critical Batch Size } b^\star & \text{if } N'(b^\star) = 0 \\ \text{Small Batch Size} & \text{if } N'(b) > 0 \text{ for all } b > 0. \end{cases} \quad (1)$$

Then, we have that $N(b^*) \geq \inf\{N\colon \min_{k\in[0:K-1]} \mathbb{E}[\|\nabla f(\boldsymbol{\theta}_k)\|] \leq \epsilon\}$, where SGD using the batch size $b^*$ generates $(\boldsymbol{\theta}_k)_{k=0}^{K-1}$.

### 1.3.3 ITERATION AND SFO COMPLEXITIES

Let $\mathcal{F}(n, \Delta, L)$ be an $L$–smooth function class with $f := \frac{1}{n}\sum_{i=1}^n f_i$ and $f(\boldsymbol{\theta}_0) - f_\star \leq \Delta$ (see (C1)) and let $\mathcal{O}(b, \sigma^2)$ be a stochastic first-order oracle class (see (C2) and (C3)). The iteration complexity $\mathcal{K}_\epsilon$ (Arjevani et al., 2023, (7)) and the SFO complexity $\mathcal{N}_\epsilon$ of SGD generating $\boldsymbol{\theta}_k(f, \mathsf{O}) = \boldsymbol{\theta}_k$ ($f \in \mathcal{F}(n, \Delta, L), \mathsf{O} \in \mathcal{O}(b, \sigma^2)$) needed to achieve an $\epsilon$–approximation are defined by

$$\mathcal{K}_\epsilon(n, b, \Delta, L, \sigma^2) := \sup_{\mathsf{O}\in\mathcal{O}(b,\sigma^2)} \sup_{f\in\mathcal{F}(n,\Delta,L)} \inf\left\{ K\colon \min_{k\in[0:K-1]} \mathbb{E}[\|\nabla f(\boldsymbol{\theta}_k)\|] \leq \epsilon \right\},$$

$$\mathcal{N}_\epsilon(n, b, \Delta, L, \sigma^2) := \sup_{\mathsf{O}\in\mathcal{O}(b,\sigma^2)} \sup_{f\in\mathcal{F}(n,\Delta,L)} \inf\left\{ N\colon \min_{k\in[0:K-1]} \mathbb{E}[\|\nabla f(\boldsymbol{\theta}_k)\|] \leq \epsilon \right\}. \quad (2)$$

Table 2 summarizes the iteration and SFO complexities (see also Theorem 3.4). It indicates that using a step-decay learning rate reduces the iteration and SFO complexities. However, since the positive constants, such as $C_i$ and $D_i$, depend on the learning rate, we need to compare numerically the performances of SGD using constant/decay learning rates. Moreover, we also need to compare the existing first-order optimizers with SGD using a step-decay learning rate to verify its usefulness. Section 4 describes numerical comparisons showing that SGD using a step-decay learning rate and small batch size performs better than the existing first-order optimizers.

Table 2: Iteration and SFO complexities of SGD using constant/decay learning rates needed to achieve an $\epsilon$–approximation (The optimal batch sizes defined as in (1) are used to compute $\mathcal{N}_\epsilon$)

| Learning Rate | | Iteration Complexity $\mathcal{K}_\epsilon$ | SFO Complexity $\mathcal{N}_\epsilon(n, b^*, \Delta, L, \sigma^2)$ |
|---|---|---|---|
| Constant $\alpha \in (0, \frac{2}{L})$ | | $O\left(\dfrac{1}{\epsilon^2}\right) = \sup\limits_{f,\mathsf{O}} K(b)$ | $O\left(\dfrac{1}{\epsilon^4}\right) = \sup\limits_{f,\mathsf{O}} \dfrac{4C_1 C_2}{\epsilon^4}$ |
| Decay $\alpha_k = \frac{1}{(k+1)^a}$ | $a \in (0, \frac{1}{2})$ | $O\left(\dfrac{1}{\epsilon^{\frac{2}{a}}}\right) = \sup\limits_{f,\mathsf{O}} K(b)$ | $O\left(\dfrac{1}{\epsilon^{\frac{2}{a}}}\right) = \sup\limits_{f,\mathsf{O}} \dfrac{(1-a)^{1-\frac{1}{a}} D_2}{a(1-2a)D_1^{1-\frac{1}{a}} \epsilon^{\frac{2}{a}}}$ |
| | $a = \frac{1}{2}$ | $O\left(\dfrac{1}{\epsilon^4}\right) = \sup\limits_{f,\mathsf{O}} K(b)$ | $O\left(\dfrac{1}{\epsilon^4}\right) = \sup\limits_{f,\mathsf{O}} \left(\dfrac{D_1 + D_2}{\epsilon^2 - D_2}\right)^2$ |
| | $a \in (\frac{1}{2}, 1)$ | $O\left(\dfrac{1}{\epsilon^{\frac{2}{1-a}}}\right) = \sup\limits_{f,\mathsf{O}} K(b)$ | $O\left(\dfrac{1}{\epsilon^{\frac{2}{1-a}}}\right) = \sup\limits_{f,\mathsf{O}} \dfrac{2a^{2-\frac{1}{1-a}}(1-a)^{-1} D_2}{(2a-1)D_1^{1-\frac{1}{1-a}} \epsilon^{\frac{2}{1-a}}}$ |
| Step Decay $\alpha_k \geq \underline{\alpha}$ | | $O\left(\dfrac{1}{\epsilon^2}\right) = \sup\limits_{f,\mathsf{O}} K(b)$ | $O\left(\dfrac{1}{\epsilon^2}\right) = \sup\limits_{f,\mathsf{O}} \dfrac{D_1 + D_3}{\underline{\alpha}\epsilon^2}$ |

## 2 NONCONVEX OPTIMIZATION AND SGD

### 2.1 NONCONVEX OPTIMIZATION IN DEEP LEARNING

Let $\mathbb{R}^d$ be a $d$-dimensional Euclidean space with inner product $\langle \boldsymbol{x}, \boldsymbol{y} \rangle := \boldsymbol{x}^\top \boldsymbol{y}$ inducing the norm $\|\boldsymbol{x}\|$ and $\mathbb{N}$ be the set of nonnegative integers. Define $[0 : n] := \{0, 1, \ldots, n\}$ for $n \geq 1$. Let $(x_k)_{k\in\mathbb{N}}$ and $(y_k)_{k\in\mathbb{N}}$ be positive real sequences and let $x(\epsilon), y(\epsilon) > 0$, where $\epsilon > 0$. $O$ denotes Landau's symbol; i.e., $y_k = O(x_k)$ if there exist $c > 0$ and $k_0 \in \mathbb{N}$ such that $y_k \leq cx_k$ for all $k \geq k_0$, and $y(\epsilon) = O(x(\epsilon))$ if there exists $c > 0$ such that $y(\epsilon) \leq cx(\epsilon)$. Given a parameter $\boldsymbol{\theta} \in \mathbb{R}^d$ and a data point $z$ in a data domain $Z$, a machine learning model provides a prediction whose quality is measured by a differentiable nonconvex loss function $\ell(\boldsymbol{\theta}; z)$. We aim to minimize the empirical loss defined for all $\boldsymbol{\theta} \in \mathbb{R}^d$ by $f(\boldsymbol{\theta}) = \frac{1}{n}\sum_{i=1}^n \ell(\boldsymbol{\theta}; z_i) = \frac{1}{n}\sum_{i=1}^n f_i(\boldsymbol{\theta})$, where $S = (z_1, z_2, \ldots, z_n)$ denotes the training set and $f_i(\cdot) := \ell(\cdot; z_i)$ denotes the loss function corresponding to the $i$-th training data $z_i$.

### 2.2 SGD

#### 2.2.1 CONDITIONS AND ALGORITHM

We assume that a stochastic first-order oracle (SFO) exists such that, for a given $\boldsymbol{\theta} \in \mathbb{R}^d$, it returns a stochastic gradient $\mathsf{G}_\xi(\boldsymbol{\theta})$ of the function $f$, where a random variable $\xi$ is independent of $\boldsymbol{\theta}$. Let $\mathbb{E}_\xi[\cdot]$ be the expectation taken with respect to $\xi$. The following are standard conditions.

(C1) $f := \frac{1}{n}\sum_{i=1}^n f_i \colon \mathbb{R}^d \to \mathbb{R}$ is $L$–smooth, i.e., $\nabla f \colon \mathbb{R}^d \to \mathbb{R}^d$ is $L$–Lipschitz continuous (i.e., $\|\nabla f(\boldsymbol{x}) - \nabla f(\boldsymbol{y})\| \leq L\|\boldsymbol{x} - \boldsymbol{y}\|$). $f$ is bounded below from $f_\star \in \mathbb{R}$. Let $\Delta > 0$ satisfy $f(\boldsymbol{\theta}_0) - f_\star \leq \Delta$, where $\boldsymbol{\theta}_0$ is an initial point.

(C2) Let $(\boldsymbol{\theta}_k)_{k\in\mathbb{N}} \subset \mathbb{R}^d$ be the sequence generated by SGD. For each iteration $k$, $\mathbb{E}_{\xi_k}[\mathsf{G}_{\xi_k}(\boldsymbol{\theta}_k)] = \nabla f(\boldsymbol{\theta}_k)$, where $\xi_0, \xi_1, \ldots$ are independent samples and the random variable $\xi_k$ is independent of $(\boldsymbol{\theta}_l)_{l=0}^k$. There exists a nonnegative constant $\sigma^2$ such that $\mathbb{E}_{\xi_k}[\|\mathsf{G}_{\xi_k}(\boldsymbol{\theta}_k) - \nabla f(\boldsymbol{\theta}_k)\|^2] \leq \sigma^2$.

(C3) For each iteration $k$, SGD samples a batch $B_k$ of size $b$ independently of $k$ and estimates the full gradient $\nabla f$ as $\nabla f_{B_k}(\boldsymbol{\theta}_k) := \frac{1}{b}\sum_{i \in [b]} \mathsf{G}_{\xi_{k,i}}(\boldsymbol{\theta}_k)$, where $\xi_{k,i}$ is a random variable generated by the $i$-th sampling in the $k$-th iteration.

Algorithm 1 is the SGD optimizer under (C1)–(C3).

---

**Algorithm 1** SGD

---

**Require:** $\alpha_k \in (0, +\infty)$ (learning rate), $b \geq 1$ (batch size), $K \geq 1$ (iteration)
**Ensure:** $\boldsymbol{\theta}_K$
1: $k \leftarrow 0, \boldsymbol{\theta}_0 \in \mathbb{R}^d$
2: **loop**
3: $\quad \nabla f_{B_k}(\boldsymbol{\theta}_k) := \frac{1}{b} \sum_{i \in [b]} \mathsf{G}_{\xi_{k,i}}(\boldsymbol{\theta}_k)$
4: $\quad \boldsymbol{\theta}_{k+1} := \boldsymbol{\theta}_k - \alpha_k \nabla f_{B_k}(\boldsymbol{\theta}_k)$
5: $\quad k \leftarrow k + 1$
6: **end loop**

---

### 2.2.2 LEARNING RATES

We use the following learning rates:

**(Constant)** $\alpha_k$ does not depend on $k \in \mathbb{N}$, i.e., $\alpha_k = \alpha < \frac{2}{L}$ ($k \in \mathbb{N}$), where the upper bound $\frac{2}{L}$ of $\alpha$ is needed to analyze SGD (see Appendix A.2).

**(Decay)** $(\alpha_k)_{k \in \mathbb{N}} \subset (0, +\infty)$ is monotone decreasing for $k$ (i.e., $\alpha_k \geq \alpha_{k+1}$) and converges to 0. In particular, we use $\alpha_k = \frac{1}{(k+1)^a}$, where **(Decay 1)** $a \in (0, \frac{1}{2}) \vee$ **(Decay 2)** $a = \frac{1}{2} \vee$ **(Decay 3)** $a \in (\frac{1}{2}, 1)$. It is guaranteed that there exists $k_0 \in \mathbb{N}$ such that, for all $k \geq k_0$, $\alpha_k < \frac{2}{L}$. We assume that $k_0 = 0$, since we can replace $\alpha_k = \frac{1}{(k+1)^a}$ with $\frac{\alpha}{(k+1)^a} \leq \alpha < \frac{2}{L}$ ($k \in \mathbb{N}$), where $\alpha \in (0, \frac{2}{L})$ is defined as in **(Constant)**.

**(Step Decay)** Let $\alpha > 0$, $\eta \in (0,1)$, $T, P \geq 1$, and $K = TP$. A step-decay learning rate is

$$\textbf{(Decay 4)} \quad (\alpha_k)_{k=0}^{K-1} = (\underbrace{\alpha, \alpha, \cdots, \alpha}_{T}, \underbrace{\alpha\eta, \alpha\eta, \cdots, \alpha\eta}_{T}, \cdots, \underbrace{\alpha\eta^{P-1}, \alpha\eta^{P-1}, \cdots, \alpha\eta^{P-1}}_{T}),$$

which is monotone decreasing for $k$. Let $\underline{\alpha} > 0$ be a lower bound of $\alpha_{K-1}$. We assume that $\alpha < \frac{2}{L}$, which implies that, for all $k \in [0 : K-1]$, $\alpha_k < \frac{2}{L}$.

## 3 OUR RESULTS

### 3.1 UPPER BOUND OF THE SQUARED NORM OF THE FULL GRADIENT

We give an upper bound of $\min_{k \in [0:K-1]} \mathbb{E}[\|\nabla f(\boldsymbol{\theta}_k)\|^2]$, where $\mathbb{E}[\cdot]$ stands for the total expectation, for the sequence generated by SGD using each of the learning rates defined in Section 2.2.2.

**Theorem 3.1 (Upper bound of the squared norm of the full gradient)** *The sequence $(\boldsymbol{\theta}_k)_{k \in \mathbb{N}}$ generated by Algorithm 1 under (C1)–(C3) satisfies that, for all $K \geq 1$,*

$$\min_{k \in [0:K-1]} \mathbb{E}\left[\|\nabla f(\boldsymbol{\theta}_k)\|^2\right] \leq \begin{cases} \dfrac{C_1}{K} + \dfrac{C_2}{b} & \textbf{(Constant)} \\[2mm] \dfrac{D_1}{K^a} + \dfrac{D_2}{(1-2a)K^a b} & \textbf{(Decay 1)} \\[2mm] \dfrac{D_1}{\sqrt{K}} + \left(\dfrac{1}{\sqrt{K}} + 1\right)\dfrac{D_2}{b} & \textbf{(Decay 2)} \\[2mm] \dfrac{D_1}{K^{1-a}} + \dfrac{2aD_2}{(2a-1)K^{1-a}b} & \textbf{(Decay 3)} \\[2mm] \dfrac{D_1}{\underline{\alpha}K} + \dfrac{D_3}{\underline{\alpha}Kb} & \textbf{(Decay 4)} \end{cases}$$

*where*

$$C_1 := \frac{2(f(\boldsymbol{\theta}_0) - f_\star)}{(2 - L\alpha)\alpha}, \quad C_2 := \frac{L\sigma^2\alpha}{2 - L\alpha},$$

$$D_1 := \begin{cases} \dfrac{2(f(\boldsymbol{\theta}_0) - f_\star)}{2 - L\alpha_0} & \textbf{(Decay 1)–(Decay 3)} \\[2mm] \dfrac{2(f(\boldsymbol{\theta}_0) - f_\star)}{2 - L\alpha} & \textbf{(Decay 4)}, \end{cases} \quad D_2 := \frac{L\sigma^2}{2 - L\alpha_0}, \quad D_3 := \frac{L\alpha^2 T\sigma^2}{(1 - \eta^2)(2 - L\alpha)}.$$

Theorem 3.1 indicates that the upper bound of $\min_{k\in[0:K-1]} \mathbb{E}[\|\nabla f(\boldsymbol{\theta}_k)\|^2]$ consists of a bias term including $f(\boldsymbol{\theta}_0) - f_\star$ and a variance term including $\sigma^2$ and that these terms become small when the number of iterations and the batch size are large. In particular, the bias term using **(Constant)** or **(Decay 4)** is $O(\frac{1}{K})$, which is a better rate than using **(Decay 1)**–**(Decay 3)**. Moreover, the variance term using **(Decay 4)** is $O(\frac{1}{Kb})$, which is a better rate than using other learning rates.

### 3.2 NUMBER OF ITERATIONS NEEDED TO ACHIEVE $\epsilon$–APPROXIMATION OF SGD

Let us consider an $\epsilon$–approximation of SGD defined as follows:
$$\mathbb{E}\left[\|\nabla f(\boldsymbol{\theta}_{K^*})\|^2\right] := \min_{k\in[0:K-1]} \mathbb{E}\left[\|\nabla f(\boldsymbol{\theta}_k)\|^2\right] \leq \epsilon^2, \tag{3}$$
where $\epsilon > 0$ is the precision and $K^* \in [0 : K - 1]$. Condition (3) implies that $\mathbb{E}[\|\nabla f(\boldsymbol{\theta}_{K^*})\|] \leq \epsilon$. Theorem 3.1 below gives the number of iterations needed to achieve an $\epsilon$–approximation (3) of SGD.

**Theorem 3.2 (Numbers of iterations needed for nonconvex optimization of SGD)** *Let $(\boldsymbol{\theta}_k)_{k\in\mathbb{N}}$ be the sequence generated by Algorithm 1 under (C1)–(C3) and let $K\colon \mathbb{R} \to \mathbb{R}$ be*

$$K(b) = \begin{cases} \dfrac{C_1 b}{\epsilon^2 b - C_2} & \textbf{(Constant)} \\[2mm] \left\{\dfrac{1}{\epsilon^2}\left(\dfrac{D_2}{(1-2a)b} + D_1\right)\right\}^{\frac{1}{a}} & \textbf{(Decay 1)} \\[2mm] \left(\dfrac{D_1 b + D_2}{\epsilon^2 b - D_2}\right)^2 & \textbf{(Decay 2)} \\[2mm] \left\{\dfrac{1}{\epsilon^2}\left(\dfrac{2aD_2}{(2a-1)b} + D_1\right)\right\}^{\frac{1}{1-a}} & \textbf{(Decay 3)} \\[2mm] \dfrac{1}{\alpha\epsilon^2}\left(\dfrac{D_3}{b} + D_1\right) & \textbf{(Decay 4)} \end{cases}$$

*where $C_1$, $C_2$, $D_1$, $D_2$, and $D_3$ are defined as in Theorem 3.1, the domain of $K$ in **(Constant)** is $b > \frac{C_2}{\epsilon^2}$, and the domain of $K$ in **(Decay 2)** is $b > \frac{D_2}{\epsilon^2}$. Then, we have the following:*

(i) *The above $K$ achieves an $\epsilon$–approximation (3).*

(ii) *The above $K$ is a monotone decreasing and convex function with respect to the batch size $b$.*

Theorem 3.2 indicates that the number of iterations needed for SGD using constant/decay learning rates to be an $\epsilon$–approximation is small when the batch size is large. Hence, it is appropriate to set a large batch size in the sense of minimizing the iterations needed for an $\epsilon$–approximation (3). However, the SFO complexity, which is the stochastic gradient computation cost, becomes larger as $b$ grows. Hence, the appropriate batch size should also minimize the SFO complexity.

### 3.3 SFO COMPLEXITY TO ACHIEVE $\epsilon$–APPROXIMATION OF SGD

Theorem 3.2 leads to the following theorem on the properties of the SFO complexity $N$ needed to achieve an $\epsilon$–approximation (3) of SGD.

**Theorem 3.3 (SFO complexity needed for nonconvex optimization of SGD)** *Let $(\boldsymbol{\theta}_k)_{k\in\mathbb{N}}$ be the sequence generated by Algorithm 1 under (C1)–(C3) and define $N\colon \mathbb{R} \to \mathbb{R}$ by*

$$N(b) = K(b)b = \begin{cases} \dfrac{C_1 b^2}{\epsilon^2 b - C_2} & \textbf{(Constant)} \\[2mm] \left\{\dfrac{1}{\epsilon^2}\left(\dfrac{D_2}{(1-2a)b} + D_1\right)\right\}^{\frac{1}{a}} b & \textbf{(Decay 1)} \\[2mm] \left(\dfrac{D_1 b + D_2}{\epsilon^2 b - D_2}\right)^2 b & \textbf{(Decay 2)} \\[2mm] \left\{\dfrac{1}{\epsilon^2}\left(\dfrac{2aD_2}{(2a-1)b} + D_1\right)\right\}^{\frac{1}{1-a}} b & \textbf{(Decay 3)} \\[2mm] \dfrac{1}{\alpha\epsilon^2}\left(D_3 + D_1 b\right) & \textbf{(Decay 4)} \end{cases}$$

*where $C_1$, $C_2$, $D_1$, $D_2$, and $D_3$ are as in Theorem 3.1, the domain of $N$ in* **(Constant)** *is $b > \frac{C_2}{\epsilon^2}$, and the domain of $N$ in* **(Decay 2)** *is $b > \frac{D_2}{\epsilon^2}$. Then, we have the following:*

(i) *The above $N$ is convex with respect to the batch size $b$.*

(ii) *There exists a critical batch size*

$$
b^\star = \begin{cases} \dfrac{2C_2}{\epsilon^2} & \textbf{(Constant)} \\[2mm] \dfrac{(1-a)D_2}{a(1-2a)D_1} & \textbf{(Decay 1)} \\[2mm] \dfrac{2a^2 D_2}{(1-a)(2a-1)D_1} & \textbf{(Decay 3)} \end{cases} \tag{4}
$$

*satisfying $N'(b^\star) = 0$ such that $b^\star$ minimizes the SFO complexity $N$.*

(iii) *For* **(Decay 2)** *and* **(Decay 4)***, $N'(b) > 0$ holds for all $b > 0$.*

Theorem 3.3(ii) indicates that, if we can set a critical batch size (4) for each of **(Constant)**, **(Decay 1)**, and **(Decay 3)**, then the SFO complexity will be minimized. However, it would be difficult to set $b^\star$ in (4) before implementing SGD, since $b^\star$ in (4) involves unknown parameters, such as $L$ and $\sigma^2$ (see Theorem 3.1 for the definitions of $C_2$, $D_1$, and $D_2$). Meanwhile, Theorem 3.3(iii) indicates that small batch sizes are appropriate when using **(Decay 2)** and **(Decay 4)** in the sense of minimizing the SFO complexity $N$.

## 3.4 Iteration and SFO complexities of SGD

Theorems 3.2 and 3.3 lead to the following theorem indicating the iteration and SFO complexities needed to achieve $\epsilon$–approximation of SGD (see also Table 2).

**Theorem 3.4 (Iteration and SFO complexities of SGD)** *The iteration and SFO complexities such that Algorithm 1 under (C1)–(C3) can be an $\epsilon$–approximation (3) are as follows:*

$$
(\mathcal{K}_\epsilon(n,b,\Delta,L,\sigma^2), \mathcal{N}_\epsilon(n,b^*,\Delta,L,\sigma^2)) = \begin{cases} \left(O\left(\dfrac{1}{\epsilon^2}\right), O\left(\dfrac{1}{\epsilon^4}\right)\right) & \textbf{(Constant)} \\[2mm] \left(O\left(\dfrac{1}{\epsilon^{\frac{2}{a}}}\right), O\left(\dfrac{1}{\epsilon^{\frac{2}{a}}}\right)\right) & \textbf{(Decay 1)} \\[2mm] \left(O\left(\dfrac{1}{\epsilon^4}\right), O\left(\dfrac{1}{\epsilon^4}\right)\right) & \textbf{(Decay 2)} \\[2mm] \left(O\left(\dfrac{1}{\epsilon^{\frac{2}{1-a}}}\right), O\left(\dfrac{1}{\epsilon^{\frac{2}{1-a}}}\right)\right) & \textbf{(Decay 3)} \\[2mm] \left(O\left(\dfrac{1}{\epsilon^2}\right), O\left(\dfrac{1}{\epsilon^2}\right)\right) & \textbf{(Decay 4)} \end{cases}
$$

*where $\mathcal{K}_\epsilon(n,b,\Delta,L,\sigma^2)$ and $\mathcal{N}_\epsilon(n,b,\Delta,L,\sigma^2)$ are defined as in (2), the optimal batch sizes (1) are used to compute $\mathcal{N}_\epsilon(n,b^*,\Delta,L,\sigma^2)$ (see also (4)), and we assume that, for* **(Constant)** *and* **(Decay 2)***, there exists $M > 0$ such that $\epsilon^2 b - C_2, \epsilon^2 b - D_2 \geq M\epsilon^2 b$ to compute $\mathcal{K}_\epsilon(n,b,\Delta,L,\sigma^2)$.*

Theorem 3.4 indicates that the iteration complexities for **(Constant)** and **(Decay 4)** are better than those for **(Decay 1)**–**(Decay 3)** and the SFO complexity for **(Decay 4)** is the best. Therefore, we can conclude that using the step-decay learning rate **(Decay 4)** is useful for SGD in the sense of minimizing the iteration and SFO complexities needed to achieve an $\epsilon$–approximation.

## 4 Numerical Results

We numerically verified the number of iterations and SFO complexities needed to achieve high test accuracy for different batch sizes in training ResNet (Appendix A.6 provides the number of iterations and SFO complexities needed to achieve high training accuracy). The parameter $\alpha$ used in **(Constant)** was determined by conducting a grid search of $\{0.001, 0.005, 0.01, 0.05, 0.1, 0.5\}$. The parameters $\alpha$ used in the decaying learning rate **(Decay 1)**–**(Decay 3)** defined by $\alpha_k = \frac{\alpha}{(k+1)^a}$ were

determined by a grid search of $\{0.001, 0.005, 0.01, 0.05, 0.1, 0.5, 1.0\}$. The parameters $\alpha$ and $\eta$ used in **(Decay 4)** were determined by a grid search of $\alpha \in \{0.125, 0.25, 0.5\}$ and $\eta \in \{0.25, 0.5, 0.75\}$. The parameter $T$ in **(Decay 4)** was set to $T = 20$ epochs. The parameter $a$ in **(Decay 1)** and **(Decay 3)** was set to $a = \frac{1}{4}$ and $a = \frac{3}{4}$, respectively. We compared SGD using **(Decay 4)** with SGD with momentum (momentum), Adam, AdamW, and RMSProp. The learning rates and hyperparameters of the four optimizers were determined on the basis of the previous results (Kingma & Ba, 2015; Loshchilov & Hutter, 2019; Tieleman & Hinton, 2012) (The weight decay used in the momentum was $5 \times 10^{-4}$). The experimental environment consisted of an NVIDIA DGX A100×8GPU and Dual AMD Rome7742 2.25-GHz, 128 Cores×2CPU. The software environment was Python 3.10.6, PyTorch 1.13.1, and CUDA 11.6. The code is available at https://anonymous.4open.science/r/SGD_with_decaying/.

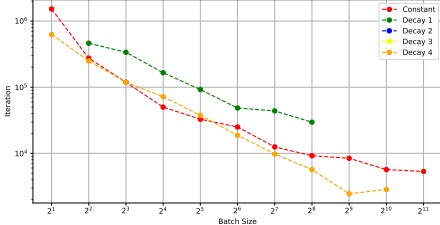

Figure 1: Number of iterations needed for SGD with (Constant), (Decay 1), (Decay 2), (Decay 3), and (Decay 4) to achieve a test accuracy of $0.9$ versus batch size (ResNet-18 on CIFAR-10)

Figure 2: SFO complexity needed for SGD with (Constant), (Decay 1), (Decay 2), (Decay 3), and (Decay 4) to achieve a test accuracy of $0.9$ versus batch size (ResNet-18 on CIFAR-10)

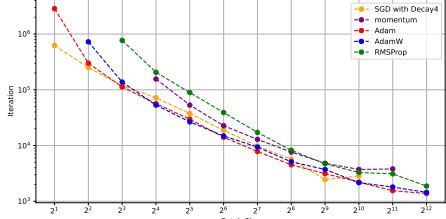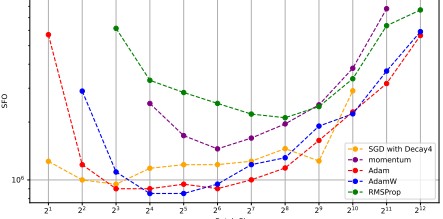

Figure 3: Number of iterations needed for SGD with (Decay 4), momentum, Adam, AdamW, and RMSProp to achieve a test accuracy of $0.9$ versus batch size (ResNet-18 on CIFAR-10)

Figure 4: SFO complexity needed for SGD with (Decay 4), momentum, Adam, AdamW, and RMSProp to achieve a test accuracy of $0.9$ versus batch size (ResNet-18 on CIFAR-10)

First, we trained ResNet-18 on CIFAR-10 dataset. The stopping condition of the optimizers was 200 epochs. Figures 1 and 2 show performance measures for five different learning rates in achieving a test accuracy of $0.9$. Figure 1 indicates that using **(Decay 2)** and **(Decay 3)** did not reach the test accuracy $0.9$ before the stopping condition was reached (Figures 9 and 10 in Appendix A.6 indicate that using **(Decay 2)** and **(Decay 3)** reached the training accuracy $0.9$). Meanwhile, Figure 1 indicates that using **(Constant)**, **(Decay 1)**, and **(Decay 4)** decreased the number of iterations. Figure 2 indicates that, in the case of SGD using **(Constant)**, a critical batch size $b^\star = 2^4$ exists at which the SFO complexity is minimized. Figures 1 and 2 indicate that, for using a small batch size $(b = 2^1, 2^2)$, SGD using **(Decay 4)** performs better than SGD using **(Constant)** and **(Decay 1)**.

Figures 3 and 4 compare SGD with **(Decay 4)** with other optimizers. These figures indicate that, for using a small batch size $(b = 2^1, 2^2)$, SGD with **(Decay 4)** performed better than the other optimizers in minimizing the number of iterations and the SFO complexity. Figure 4 also indicates that the existing optimizers using constant learning rates had critical batch sizes minimizing the SFO complexities. In particular, AdamW using the critical batch size $b^\star = 2^5$ (Figure 4) and SGD using **(Constant)** and $b^\star = 2^4$ (Figure 2) performed well. However, it would be difficult to set the critical batch size in advance, since it involves unknown parameters $L$ and $\sigma^2$ (see (4) and $C_2 = \frac{L\sigma^2\alpha}{2-L\alpha}$

(computing the Lipschitz constant $L$ is NP-hard (Virmaux & Scaman, 2018)). Meanwhile, we can set small batch sizes for using SGD with a step-decay learning rate.

Next, we trained ResNet-18 on the CIFAR-100 dataset. The stopping condition of the optimizers was 1000 epochs. Figures 5 and 6 show performance measures of SGD for five different learning rates in achieving a test accuracy of $0.6$. As in Figures 3 and 4, Figures 7 and 8 indicate that, for using a small batch size ($b = 2^1, 2^2, 2^3, 2^4$), SGD with (**Decay 4**) reduced the SFO complexity. Figures 7 and 8 indicate that using the existing optimizers with $b = 2^1$ did not reach the test accuracy $0.6$ before the stopping condition was reached, in contrast to SGD with (**Decay 4**) and $b = 2^1$. Moreover, the SFO complexity of SGD with (**Decay 4**) and the batch size $b = 2^4$ was the smallest of other optimizers for any batch size. Figures 5–8 indicate that SGD with (**Decay 4**) was more robust than other optimizers in terms of using small batch sizes (See Figures 17–20 for the results on the MNIST dataset).

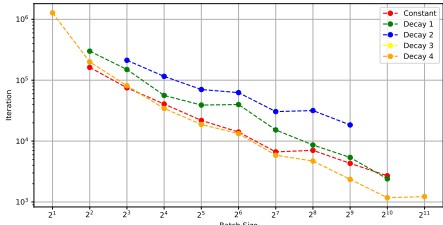

Figure 5: Number of iterations needed for SGD with (Constant), (Decay 1), (Decay 2), (Decay 3), and (Decay 4) to achieve a test accuracy of $0.6$ versus batch size (ResNet-18 on CIFAR-100)

Figure 6: SFO complexity needed for SGD with (Constant), (Decay 1), (Decay 2), (Decay 3), and (Decay 4) to achieve a test accuracy of $0.6$ versus batch size (ResNet-18 on CIFAR-100)

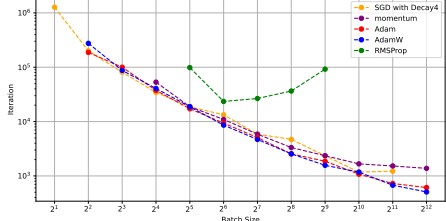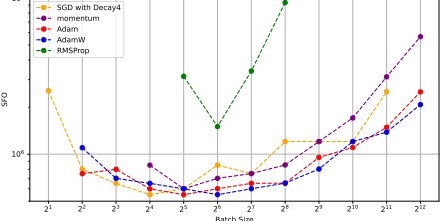

Figure 7: Number of iterations needed for SGD with (Decay 4), momentum, Adam, AdamW, and RMSProp to achieve a test accuracy of $0.6$ versus batch size (ResNet-18 on CIFAR-100)

Figure 8: SFO complexity needed for SGD with (Decay 4), momentum, Adam, AdamW, and RMSProp to achieve a test accuracy of $0.6$ versus batch size (ResNet-18 on CIFAR-100)

## 5 CONCLUSION AND FUTURE WORK

This paper investigated the required number of iterations and SFO complexities of SGD using constant/decay learning rates to achieve an $\epsilon$–approximation. Our theoretical analyses indicated that the number of iterations needed for an $\epsilon$–approximation is monotone decreasing and convex with respect to the batch size and the SFO complexity needed for an $\epsilon$–approximation is convex with respect to the batch size. Moreover, we showed that SGD using a step-decay learning rate and a small batch size reduces the SFO complexity. The numerical results indicated that SGD using a step-decay learning rate and a small batch size performs better than the existing optimizers in the sense of minimizing the SFO complexity.

The results in this paper can be applied to only SGD. This is a limitation of our work. Hence, in the future, we should investigate whether our results can be applied to variants of SGD, such as the momentum methods and adaptive methods.

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

## A    APPENDIX

### A.1    LEMMA

First, we will prove the following lemma.

**Lemma A.1** *The sequence* $(\boldsymbol{\theta}_k)_{k\in\mathbb{N}}$ *generated by Algorithm 1 under (C1)–(C3) satisfies that, for all* $K \geq 1$,

$$\sum_{k=0}^{K-1} \alpha_k \left( 1 - \frac{L\alpha_k}{2} \right) \mathbb{E}\left[ \|\nabla f(\boldsymbol{\theta}_k)\|^2 \right] \leq \mathbb{E}\left[ f(\boldsymbol{\theta}_0) - f_\star \right] + \frac{L\sigma^2}{2b} \sum_{k=0}^{K-1} \alpha_k^2.$$

*Proof:* Condition (C1) (*L*-smoothness of $f$) implies that the descent lemma holds, i.e., for all $k \in \mathbb{N}$,

$$f(\boldsymbol{\theta}_{k+1}) \leq f(\boldsymbol{\theta}_k) + \langle \nabla f(\boldsymbol{\theta}_k), \boldsymbol{\theta}_{k+1} - \boldsymbol{\theta}_k \rangle + \frac{L}{2} \|\boldsymbol{\theta}_{k+1} - \boldsymbol{\theta}_k\|^2,$$

which, together with $\boldsymbol{\theta}_{k+1} := \boldsymbol{\theta}_k - \alpha_k \nabla f_{B_k}(\boldsymbol{\theta}_k)$, implies that

$$f(\boldsymbol{\theta}_{k+1}) \leq f(\boldsymbol{\theta}_k) - \alpha_k \langle \nabla f(\boldsymbol{\theta}_k), \nabla f_{B_k}(\boldsymbol{\theta}_k) \rangle + \frac{L\alpha_k^2}{2} \|\nabla f_{B_k}(\boldsymbol{\theta}_k)\|^2. \tag{5}$$

Condition (C2) guarantees that

$$\mathbb{E}\left[ \nabla f_{B_k}(\boldsymbol{\theta}_k) | \boldsymbol{\theta}_k \right] = \nabla f(\boldsymbol{\theta}_k) \text{ and } \mathbb{E}\left[ \|\nabla f_{B_k}(\boldsymbol{\theta}_k) - \nabla f(\boldsymbol{\theta}_k)\|^2 | \boldsymbol{\theta}_k \right] \leq \frac{\sigma^2}{b}. \tag{6}$$

Hence, we have

$$\begin{aligned}
\mathbb{E}\left[ \|\nabla f_{B_k}(\boldsymbol{\theta}_k)\|^2 | \boldsymbol{\theta}_k \right] &= \mathbb{E}\left[ \|\nabla f_{B_k}(\boldsymbol{\theta}_k) - \nabla f(\boldsymbol{\theta}_k) + \nabla f(\boldsymbol{\theta}_k)\|^2 | \boldsymbol{\theta}_k \right] \\
&= \mathbb{E}\left[ \|\nabla f_{B_k}(\boldsymbol{\theta}_k) - \nabla f(\boldsymbol{\theta}_k)\|^2 | \boldsymbol{\theta}_k \right] + 2\mathbb{E}\left[ \langle \nabla f_{B_k}(\boldsymbol{\theta}_k) - \nabla f(\boldsymbol{\theta}_k), \nabla f(\boldsymbol{\theta}_k) \rangle | \boldsymbol{\theta}_k \right] \\
&\quad + \mathbb{E}\left[ \|\nabla f(\boldsymbol{\theta}_k)\|^2 | \boldsymbol{\theta}_k \right] \\
&\leq \frac{\sigma^2}{b} + \mathbb{E}\left[ \|\nabla f(\boldsymbol{\theta}_k)\|^2 \right].
\end{aligned} \tag{7}$$

Taking the expectation conditioned on $\boldsymbol{\theta}_k$ on both sides of (5), together with (6) and (7), guarantees that, for all $k \in \mathbb{N}$,

$$\begin{aligned}
\mathbb{E}\left[ f(\boldsymbol{\theta}_{k+1}) | \boldsymbol{\theta}_k \right] &\leq f(\boldsymbol{\theta}_k) - \alpha_k \mathbb{E}\left[ \langle \nabla f(\boldsymbol{\theta}_k), \nabla f_{B_k}(\boldsymbol{\theta}_k) \rangle | \boldsymbol{\theta}_k \right] + \frac{L\alpha_k^2}{2} \mathbb{E}\left[ \|\nabla f_{B_k}(\boldsymbol{\theta}_k)\|^2 | \boldsymbol{\theta}_k \right] \\
&\leq f(\boldsymbol{\theta}_k) - \alpha_k \|\nabla f(\boldsymbol{\theta}_k)\|^2 + \frac{L\alpha_k^2}{2} \left( \frac{\sigma^2}{b} + \|\nabla f(\boldsymbol{\theta}_k)\|^2 \right).
\end{aligned}$$

Hence, taking the total expectation on both sides of the above inequality ensures that, for all $k \in \mathbb{N}$,

$$\alpha_k \left(1 - \frac{L\alpha_k}{2}\right) \mathbb{E}\left[\|\nabla f(\boldsymbol{\theta}_k)\|^2\right] \leq \mathbb{E}\left[f(\boldsymbol{\theta}_k) - f(\boldsymbol{\theta}_{k+1})\right] + \frac{L\sigma^2\alpha_k^2}{2b}.$$

Let $K \geq 1$. Summing the above inequality from $k = 0$ to $k = K - 1$ ensures that

$$\sum_{k=0}^{K-1} \alpha_k \left(1 - \frac{L\alpha_k}{2}\right) \mathbb{E}\left[\|\nabla f(\boldsymbol{\theta}_k)\|^2\right] \leq \mathbb{E}\left[f(\boldsymbol{\theta}_0) - f(\boldsymbol{\theta}_K)\right] + \frac{L\sigma^2}{2b} \sum_{k=0}^{K-1} \alpha_k^2,$$

which, together with (C1) (the lower bound $f_\star$ of $f$), implies that the assertion in Lemma A.1 holds.
□

## A.2 PROOF OF THEOREM 3.1

**(Constant):** Lemma A.1 with $\alpha_k = \alpha$ implies that

$$\alpha \left(1 - \frac{L\alpha}{2}\right) \sum_{k=0}^{K-1} \mathbb{E}\left[\|\nabla f(\boldsymbol{\theta}_k)\|^2\right] \leq \mathbb{E}\left[f(\boldsymbol{\theta}_0) - f_\star\right] + \frac{L\sigma^2\alpha^2 K}{2b}.$$

Since $\alpha < \frac{2}{L}$, we have that

$$\min_{k \in [0:K-1]} \mathbb{E}\left[\|\nabla f(\boldsymbol{\theta}_k)\|^2\right] \leq \frac{1}{K} \sum_{k=0}^{K-1} \mathbb{E}\left[\|\nabla f(\boldsymbol{\theta}_k)\|^2\right] \leq \underbrace{\frac{2(f(\boldsymbol{\theta}_0) - f_\star)}{(2 - L\alpha)\alpha}}_{C_1} \frac{1}{K} + \underbrace{\frac{L\sigma^2\alpha}{2 - L\alpha}}_{C_2} \frac{1}{b}.$$

**(Decay):** Since $(\alpha_k)_{k \in \mathbb{N}}$ converges to 0, there exists $k_0 \in \mathbb{N}$ such that, for all $k \geq k_0$, $\alpha_k < \frac{2}{L}$. We assume that $k_0 = 0$ (see Section 2.2.2). Lemma A.1 ensures that, for all $K \geq 1$,

$$\sum_{k=0}^{K-1} \alpha_k \left(1 - \frac{L\alpha_k}{2}\right) \mathbb{E}\left[\|\nabla f(\boldsymbol{\theta}_k)\|^2\right] \leq \mathbb{E}\left[f(\boldsymbol{\theta}_0) - f_\star\right] + \frac{L\sigma^2}{2b} \sum_{k=0}^{K-1} \alpha_k^2,$$

which, together with $\alpha_{k+1} \leq \alpha_k < \frac{2}{L}$ ($k \in \mathbb{N}$), implies that

$$\alpha_{K-1} \left(1 - \frac{L\alpha_0}{2}\right) \sum_{k=0}^{K-1} \mathbb{E}\left[\|\nabla f(\boldsymbol{\theta}_k)\|^2\right] \leq \mathbb{E}\left[f(\boldsymbol{\theta}_0) - f_\star\right] + \frac{L\sigma^2}{2b} \sum_{k=0}^{K-1} \alpha_k^2.$$

Hence, we have that

$$\sum_{k=0}^{K-1} \mathbb{E}\left[\|\nabla f(\boldsymbol{\theta}_k)\|^2\right] \leq \frac{2(f(\boldsymbol{\theta}_0) - f_\star)}{(2 - L\alpha_0)\alpha_{K-1}} + \frac{L\sigma^2}{b(2 - L\alpha_0)\alpha_{K-1}} \sum_{k=0}^{K-1} \alpha_k^2,$$

which implies that

$$\min_{k \in [0:K-1]} \mathbb{E}\left[\|\nabla f(\boldsymbol{\theta}_k)\|^2\right] \leq \underbrace{\frac{2(f(\boldsymbol{\theta}_0) - f_\star)}{2 - L\alpha_0}}_{D_1} \frac{1}{K\alpha_{K-1}} + \frac{1}{b} \underbrace{\frac{L\sigma^2}{2 - L\alpha_0}}_{D_2} \frac{1}{K\alpha_{K-1}} \sum_{k=0}^{K-1} \alpha_k^2.$$

Meanwhile, we have that

$$\sum_{k=0}^{K-1} \alpha_k^2 = \sum_{k=0}^{K-1} \frac{1}{(k+1)^{2a}} \leq 1 + \int_0^{K-1} \frac{\mathrm{d}t}{(t+1)^{2a}} \leq \begin{cases} \dfrac{1}{1-2a}K^{1-2a} & \textbf{(Decay 1)} \\ 1 + \log K & \textbf{(Decay 2)} \\ \dfrac{2a}{2a-1} & \textbf{(Decay 3)} \end{cases}$$

and

$$\frac{1}{K\alpha_{K-1}} = \frac{1}{K^{1-a}}.$$

Accordingly, we have that

$$\min_{k \in [0:K-1]} \mathbb{E}\left[\|\nabla f(\boldsymbol{\theta}_k)\|^2\right] \leq \begin{cases} \dfrac{D_1}{K^{1-a}} + \dfrac{D_2}{(1-2a)K^a b} & \textbf{(Decay 1)} \\ \dfrac{D_1}{\sqrt{K}} + \dfrac{D_2(1+\log K)}{\sqrt{K}b} & \textbf{(Decay 2)} \\ \dfrac{D_1}{K^{1-a}} + \dfrac{2aD_2}{(2a-1)K^{1-a}b} & \textbf{(Decay 3)} \end{cases}$$

which, together with $\log K < \sqrt{K}$ and the condition on $a$, implies that

$$\min_{k \in [0:K-1]} \mathbb{E}\left[\|\nabla f(\boldsymbol{\theta}_k)\|^2\right] \leq \begin{cases} \dfrac{D_1}{K^a} + \dfrac{D_2}{(1-2a)K^a b} & \textbf{(Decay 1)} \\ \dfrac{D_1}{\sqrt{K}} + \left(\dfrac{1}{\sqrt{K}} + 1\right)\dfrac{D_2}{b} & \textbf{(Decay 2)} \\ \dfrac{D_1}{K^{1-a}} + \dfrac{2aD_2}{(2a-1)K^{1-a}b} & \textbf{(Decay 3)}. \end{cases}$$

**(Step Decay):** We have that

$$\sum_{k=0}^{K-1} \alpha_k^2 \leq \sum_{k=0}^{+\infty} \alpha_k^2 \leq \sum_{k=0}^{+\infty} \alpha^2 T \eta^{2k} = \frac{\alpha^2 T}{1 - \eta^2} \text{ and } \frac{1}{K\alpha_{K-1}} \leq \frac{1}{\underline{\alpha}K}.$$

Hence, from $\alpha_k \leq \alpha$,

$$\min_{k \in [0:K-1]} \mathbb{E}\left[\|\nabla f(\boldsymbol{\theta}_k)\|^2\right] \leq \frac{D_1}{\underline{\alpha}K} + \underbrace{\frac{L\alpha^2 T \sigma^2}{(1-\eta^2)(2-L\alpha)}}_{D_3} \frac{1}{\underline{\alpha}Kb} \quad \textbf{(Decay 4)}.$$

$\square$

### A.3 PROOF OF THEOREM 3.2

(i) Let us consider the case of **(Constant)**. We consider that the upper bound $\frac{C_1}{K} + \frac{C_2}{b}$ in Theorem 3.1 is equal to $\epsilon^2$. This implies that $K = \frac{C_1 b}{\epsilon^2 b - C_2}$ achieves an $\epsilon$–approximation. A discussion similar to the one showing that $K = \frac{C_1 b}{\epsilon^2 b - C_2}$ is an $\epsilon$–approximation ensures that the assertion in Theorem 3.2(i) is true.

(ii) It is sufficient to prove that $K' = K'(b) < 0$ and $K'' = K''(b) > 0$ hold.

**(Constant):** Let $K = \frac{C_1 b}{\epsilon^2 b - C_2}$. Then, we have that

$$K' = \frac{C_1(\epsilon^2 b - C_2) - \epsilon^2 C_1 b}{(\epsilon^2 b - C_2)^2} = -\frac{C_1 C_2}{(\epsilon^2 b - C_2)^2} < 0,$$

$$K'' = \frac{2\epsilon^2 C_1 C_2(\epsilon^2 b - C_2)}{(\epsilon^2 b - C_2)^4} = \frac{2\epsilon^2 C_1 C_2((\frac{C_1}{K} + \frac{C_2}{b})b - C_2)}{(\epsilon^2 b - C_2)^4} = \frac{2\epsilon^2 C_1^2 C_2^2}{K(\epsilon^2 b - C_2)^4} > 0.$$

**(Decay 1):** Let $K = \left(\frac{1}{\epsilon^2}\left(D_1 + \frac{D_2}{(1-2a)b}\right)\right)^{\frac{1}{a}}$. Then, we have that

$$K' = \frac{1}{a}\left\{\frac{1}{\epsilon^2}\left(D_1 + \frac{D_2}{(1-2a)b}\right)\right\}^{\frac{1}{a}-1}\left(-\frac{D_2}{\epsilon^2(1-2a)b^2}\right)$$

$$= -\frac{D_2}{a\epsilon^2(1-2a)b^2}\left\{\frac{1}{\epsilon^2}\left(D_1 + \frac{D_2}{(1-2a)b}\right)\right\}^{\frac{1-a}{a}} < 0,$$

$$K'' = \frac{2D_2}{a\epsilon^2(1-2a)b^3}\left\{\frac{1}{\epsilon^2}\left(D_1 + \frac{D_2}{(1-2a)b}\right)\right\}^{\frac{1-a}{a}}$$

$$+ \frac{2(1-a)D_2}{a^2\epsilon^2(1-2a)b^3}\left\{\frac{1}{\epsilon^2}\left(D_1 + \frac{D_2}{(1-2a)b}\right)\right\}^{\frac{1-a}{a}-1}\frac{D_2}{\epsilon^2(1-2a)b^2} > 0.$$

**(Decay 2):** Let $K = (\frac{bD_1 + D_2}{b\epsilon^2 - D_2})^2$. Then, we have that

$$K' = \frac{2D_1(bD_1 + D_2)(b\epsilon^2 - D_2)^2 - 2\epsilon^2(b\epsilon^2 - D_2)(bD_1 + D_2)^2}{(b\epsilon^2 - D_2)^4},$$

which, together with $b\epsilon^2 - D_2 > 0$, implies that

$$
\begin{aligned}
(b\epsilon^2 - D_2)^3 K' &= 2D_1(bD_1 + D_2)(b\epsilon^2 - D_2) - 2\epsilon^2(bD_1 + D_2)^2 \\
&= 2D_1(\epsilon^2 D_1 b^2 + (\epsilon^2 D_2 - D_1 D_2)b - D_2) - 2\epsilon^2(b^2 D_1^2 + 2D_1 D_2 b + D_2^2) \\
&= 2\epsilon^2 D_1^2 b^2 + 2D_1(\epsilon^2 D_2 - D_1 D_2)b - 2D_1 D_2 - 2\epsilon^2 D_1^2 b^2 - 4\epsilon^2 D_1 D_2 b - 2\epsilon^2 D_2^2 \\
&= -2D_1 D_2(D_1 + 2\epsilon^2)b - 2D_2(D_1 + \epsilon^2 D_2) < 0.
\end{aligned}
$$

Moreover,

$$K'' = \frac{-2D_1 D_2(D_1 + 2\epsilon^2)(b\epsilon^2 - D_2)^3 + 3\epsilon^2(b\epsilon^2 - D_2)^2(2D_1 D_2(D_1 + 2\epsilon^2)b + 2D_2(D_1 + \epsilon^2 D_2))}{(b\epsilon^2 - D_2)^6},$$

which implies that

$$
\begin{aligned}
&(b\epsilon^2 - D_2)^4 K'' \\
&= -2D_1 D_2(D_1 + 2\epsilon^2)(b\epsilon^2 - D_2) + 3\epsilon^2(2D_1 D_2(D_1 + 2\epsilon^2)b + 2D_2(D_1 + \epsilon^2 D_2)) \\
&= -2D_1 D_2(\epsilon^2 D_1 b - D_1 D_2 + 2\epsilon^4 b - 2\epsilon^2 D_2) + 3\epsilon^2(2D_1^2 D_2 b + 4\epsilon^2 D_1 D_2 b + 2D_1 D_2 + 2\epsilon^2 D_2^2) \\
&= -2\epsilon^2 D_1^2 D_2 b + 2D_1^2 D_2^2 - 4\epsilon^4 D_1 D_2 b + 4\epsilon^2 D_1 D_2^2 + 6\epsilon^2 D_1^2 D_2 b \\
&\quad + 12\epsilon^4 D_1 D_2 b + 6\epsilon^2 D_1 D_2 + 6\epsilon^4 D_2^2 \\
&= 4\epsilon^2 D_1 D_2(D_1 + 2\epsilon^2)b + 2D_1^2 D_2^2 + 4\epsilon^2 D_1 D_2^2 + 6\epsilon^2 D_1 D_2 + 6\epsilon^4 D_2^2 > 0.
\end{aligned}
$$

**(Decay 3):** Let $K = (\frac{1}{\epsilon^2}(D_1 + \frac{2aD_2}{(2a-1)b}))^{\frac{1}{1-a}}$. Then, we have that

$$
\begin{aligned}
K' &= \frac{1}{1-a}\left\{\frac{1}{\epsilon^2}\left(D_1 + \frac{2aD_2}{(2a-1)b}\right)\right\}^{\frac{1}{1-a}-1}\left(-\frac{2aD_2}{\epsilon^2(2a-1)b^2}\right) \\
&= -\frac{2aD_2}{\epsilon^2(1-a)(2a-1)b^2}\left\{\frac{1}{\epsilon^2}\left(D_1 + \frac{2aD_2}{(2a-1)b}\right)\right\}^{\frac{a}{1-a}} < 0, \\
K'' &= \frac{4aD_2}{\epsilon^2(1-a)(2a-1)b^3}\left\{\frac{1}{\epsilon^2}\left(D_1 + \frac{2aD_2}{(2a-1)b}\right)\right\}^{\frac{a}{1-a}} \\
&\quad + \frac{2a^2 D_2}{\epsilon^2(1-a)^2(2a-1)b^2}\left\{\frac{1}{\epsilon^2}\left(D_1 + \frac{2aD_2}{(2a-1)b}\right)\right\}^{\frac{a}{1-a}-1}\frac{2aD_2}{\epsilon^2(2a-1)b^2} > 0.
\end{aligned}
$$

**(Decay 4):** Let $K = (D_1 + \frac{D_3}{b})\frac{1}{\epsilon^2 \underline{\alpha}}$. Then, we have that

$$K' = -\frac{D_3}{\epsilon^2 \underline{\alpha} b^2} < 0, \ K'' = \frac{2D_3}{\epsilon^2 \underline{\alpha} b^3} > 0.$$

$\square$

## A.4 PROOF OF THEOREM 3.3

**(Constant):** Let $N = \frac{C_1 b^2}{\epsilon^2 b - C_2}$. Then, we have that

$$N' = \frac{2C_1 b(\epsilon^2 b - C_2) - \epsilon^2 C_1 b^2}{(\epsilon^2 b - C_2)^2} = \frac{C_1 b(\epsilon^2 b - 2C_2)}{(\epsilon^2 b - C_2)^2}.$$

If $N' = 0$, we have that $\epsilon^2 b - 2C_2 = 0$, i.e., $b = \frac{2C_2}{\epsilon^2}$. Moreover,

$$N'' = \frac{(2\epsilon^2 C_1 b - 2C_1 C_2)(\epsilon^2 b - C_2)^2 - 2\epsilon^2(\epsilon^2 b - C_2)(\epsilon^2 C_1 b^2 - 2C_1 C_2 b)}{(\epsilon^2 b - C_2)^4}$$

$$
\begin{aligned}
(\epsilon^2 b - C_2)^3 N'' &= (2\epsilon^2 C_1 b - 2C_1 C_2)(\epsilon^2 b - C_2) - 2\epsilon^2(\epsilon^2 C_1 b^2 - 2C_1 C_2 b) \\
&= 2C_1 C_2^2 > 0,
\end{aligned}
$$

which implies that $N$ is convex. Hence, there is a critical batch size $b^\star = \frac{2C_2}{\epsilon^2} > 0$ at which $N$ is minimized.

**(Decay 1):** Let $N = Kb$. Then, we have that

$$
\begin{aligned}
N' &= K + bK' \\
&= \left\{ \frac{1}{\epsilon^2} \left( D_1 + \frac{D_2}{(1-2a)b} \right) \right\}^{\frac{1}{a}} + \frac{1}{a} \left\{ \frac{1}{\epsilon^2} \left( D_1 + \frac{D_2}{(1-2a)b} \right) \right\}^{\frac{1}{a}-1} \left( -\frac{D_2}{\epsilon^2(1-2a)b^2} \right) b \\
&= \left\{ \frac{1}{\epsilon^2} \left( D_1 + \frac{D_2}{(1-2a)b} \right) \right\}^{\frac{1}{a}-1} \left\{ \frac{1}{\epsilon^2} \left( D_1 + \frac{D_2}{(1-2a)b} \right) - \frac{D_2}{a\epsilon^2(1-2a)b} \right\}.
\end{aligned}
$$

If $N' = 0$, we have that

$$
\frac{1}{\epsilon^2} \left( D_1 + \frac{D_2}{(1-2a)b} \right) - \frac{D_2}{a\epsilon^2(1-2a)b} = 0, \text{ i.e., } b = \frac{D_2(a-1)}{aD_1(2a-1)}.
$$

Moreover,

$$
\begin{aligned}
N'' &= K' + (K' + bK'') = 2K' + bK'' \\
&= -\frac{2D_2}{a\epsilon^2(1-2a)b^2} \left\{ \frac{1}{\epsilon^2} \left( D_1 + \frac{D_2}{(1-2a)b} \right) \right\}^{\frac{1-a}{a}} + \frac{2D_2}{a\epsilon^2(1-2a)b^2} \left\{ \frac{1}{\epsilon^2} \left( D_1 + \frac{D_2}{(1-2a)b} \right) \right\}^{\frac{1-a}{a}} \\
&\quad + \frac{2(1-a)D_2}{a^2\epsilon^2(1-2a)b^2} \left\{ \frac{1}{\epsilon^2} \left( D_1 + \frac{D_2}{(1-2a)b} \right) \right\}^{\frac{1-a}{a}-1} \frac{D_2}{\epsilon^2(1-2a)b^2} \\
&= \frac{2(1-a)D_2}{a^2\epsilon^2(1-2a)b^2} \left\{ \frac{1}{\epsilon^2} \left( D_1 + \frac{D_2}{(1-2a)b} \right) \right\}^{\frac{1-a}{a}-1} \frac{D_2}{\epsilon^2(1-2a)b^2} > 0,
\end{aligned}
$$

which implies that $N$ is convex. Hence, there is a critical batch size $b^\star = \frac{D_2(a-1)}{aD_1(2a-1)} > 0$.

**(Decay 2):** Let $N = bK$. Then, we have that

$$
\begin{aligned}
N' &= K + bK' \\
&= \frac{(bD_1 - D_2)^2}{(b\epsilon^2 - D_2)^2} - \frac{b(2D_1D_2(D_1 + 2\epsilon^2)b - 2D_2(D_1 + \epsilon^2 D_2))}{(b\epsilon^2 - D_2)^3} \\
&= \frac{1}{(b\epsilon^2 - D_2)^3} \{ (bD_1 - D_2)^2(b\epsilon^2 - D_2) - 2D_1D_2(D_1 + 2\epsilon^2)b^2 - 2D_2(D_1 + \epsilon^2 D_2)b \} \\
&= \frac{1}{(b\epsilon^2 - D_2)^3} \{ (D_1^2 b^2 - 2D_1D_2 b + D_2^2)(b\epsilon^2 - D_2) - 2D_1^2 D_2 b^2 - 4\epsilon^2 D_1 D_2 b^2 \\
&\quad - 2D_1 D_2 b - 2\epsilon^2 D_2^2 b \} \\
&= \frac{1}{(b\epsilon^2 - D_2)^3} (\epsilon^2 D_1^2 b^3 - D_1 D_2 b^2 - 2\epsilon^2 D_1 D_2 b^2 + 2D_1 D_2^2 b + \epsilon^2 D_2^2 b - D_2^3 \\
&\quad - 2D_1^2 D_2 b^2 - 4\epsilon^2 D_1 D_2 b^2 - 2D_1 D_2 b - 2\epsilon^2 D_2^2 b) \\
&= \frac{1}{(b\epsilon^2 - D_2)^3} (\epsilon^2 D_1^2 b^3 - D_1 D_2 b^2 - 6\epsilon^2 D_1 D_2 b^2 - 2D_1^2 D_2 b^2 + 2D_1 D_2^2 b - \epsilon^2 D_2^2 b - D_2^3) \\
&= \frac{1}{(b\epsilon^2 - D_2)^3} (D_1 b + D_2)(\epsilon^2 D_1 b^2 - D_2(3D_1 + \epsilon^2)b - D_2^2).
\end{aligned}
$$

If $N' = 0$, we have that $D_1 b + D_2 = 0$, i.e., $b = -\frac{D_2}{D_1} < 0$. Moreover,

$$
\begin{aligned}
N'' &= 2K' + bK'' \\
&= -\frac{2(2D_1 D_2 (D_1 + 2\epsilon^2) b - 2D_2 (D_1 + \epsilon^2 D_2))}{(b\epsilon^2 - D_2)^3} \\
&\quad + \frac{4\epsilon^2 D_1 D_2 (D_1 + 2\epsilon^2) b^2 + (2D_1^2 D_2^2 + 4\epsilon^2 D_1 D_2^2 + 6\epsilon^2 D_1 D_2 + 6\epsilon^4 D_2^2) b}{(b\epsilon^2 - D_2)^4} \\
&= (-2(2D_1 D_2 (D_1 + 2\epsilon^2) b - 2D_2 (D_1 + \epsilon^2 D_2))(b\epsilon^2 - D_2) \\
&\quad + 4\epsilon^2 D_1 D_2 (D_1 + 2\epsilon^2) b^2 + (2D_1^2 D_2^2 + 4\epsilon^2 D_1 D_2^2 + 6\epsilon^2 D_1 D_2 + 6\epsilon^4 D_2^2) b) \frac{1}{(b\epsilon^2 - D_2)^4} \\
&= (-4\epsilon^2 D_1^2 D_2 b^2 + 8\epsilon^4 D_1 D_2 b^2 + 4D_1^2 D_2^2 b + 8\epsilon^2 D_1 D_2^2 b - 2\epsilon^2 D_1 D_2 b - 2\epsilon^4 D_2^2 b + 2D_1 D_2^2 \\
&\quad + 2\epsilon^2 D_2^3 + 4\epsilon^2 D_1^2 D_2 b^2 + 8\epsilon^4 D_1 D_2 b^2 + 2D_1^2 D_2^2 b + 4\epsilon^2 D_1 D_2^2 b + 6\epsilon^2 D_1 D_2 b + 6\epsilon^4 D_2^2 b) \\
&\quad \times \frac{1}{(b\epsilon^2 - D_2)^4} \\
&= (4D_1^2 D_2^2 b + 12\epsilon^2 D_1 D_2^2 b + 2D_1 D_2^2 + 2\epsilon^2 D_2^3 + 2D_1^2 D_2^2 b + 4\epsilon^2 D_1 D_2 b + 4\epsilon^4 D_2^2 b) \\
&\quad \times \frac{1}{(b\epsilon^2 - D_2)^4} > 0,
\end{aligned}
$$

which implies that $N$ is convex and that a critical batch size does not exist.

**(Decay 3):** Let $N = bK$. Then, we have that

$$
\begin{aligned}
N' &= K + bK' \\
&= \left\{ \frac{1}{\epsilon^2} \left( D_1 + \frac{2aD_2}{(2a-1)b} \right) \right\}^{\frac{1}{1-a}} - \frac{2aD_2}{\epsilon^2 (1-a)(2a-1)b} \left\{ \frac{1}{\epsilon^2} \left( D_1 + \frac{2aD_2}{(2a-1)b} \right) \right\}^{\frac{1}{1-a} - 1} \\
&= \left\{ \frac{1}{\epsilon^2} \left( D_1 + \frac{2aD_2}{(2a-1)b} \right) \right\}^{\frac{1}{1-a} - 1} \left\{ \frac{1}{\epsilon^2} \left( D_1 + \frac{2aD_2}{(2a-1)b} \right) - \frac{2aD_2}{\epsilon^2 (1-a)(2a-1)b} \right\}.
\end{aligned}
$$

If $N' = 0$, we have that

$$
\frac{1}{\epsilon^2} \left( D_1 + \frac{2aD_2}{(2a-1)b} \right) - \frac{2aD_2}{\epsilon^2 (1-a)(2a-1)b} = 0, \text{ i.e., } b = \frac{2a^2 D_2}{(2a-1)(1-a)D_1}.
$$

Moreover,

$$
\begin{aligned}
N'' &= 2K' + bK'' \\
&= -\frac{2aD_2}{\epsilon^2 (1-a)(2a-1)b^2} \left\{ \frac{1}{\epsilon^2} \left( D_1 + \frac{2aD_2}{(2a-1)b} \right) \right\}^{\frac{a}{1-a}} \\
&\quad + \frac{2aD_2}{\epsilon^2 (1-a)(2a-1)b^2} \left\{ \frac{1}{\epsilon^2} \left( D_1 + \frac{2aD_2}{(2a-1)b} \right) \right\}^{\frac{a}{1-a}} \\
&\quad + \frac{2a^2 D_2}{\epsilon^2 (1-a)^2 (2a-1)b} \left\{ \frac{1}{\epsilon^2} \left( D_1 + \frac{2aD_2}{(2a-1)b} \right) \right\}^{\frac{2a-1}{1-a}} \frac{2aD_2}{\epsilon^2 (2a-1)b^2} \\
&= \frac{2a^2 D_2}{\epsilon^2 (1-a)^2 (2a-1)b} \left\{ \frac{1}{\epsilon^2 (D_1 + \frac{2aD_2}{(2a-1)b})} \right\}^{\frac{2a-1}{1-a}} \frac{2aD_2}{\epsilon^2 (2a-1)b^2} > 0,
\end{aligned}
$$

which implies that $N$ is convex. Hence, there is a critical batch size $b^\star = \frac{2a^2 D_2}{(2a-1)(1-a)D_1} > 0$.

**(Decay 4):** We have that

$$
N = bK = (D_1 b + D_3) \frac{1}{\epsilon^2 \underline{\alpha}} > 0, \ N' = \frac{D_1}{\epsilon^2 \underline{\alpha}} > 0, \ N'' = 2K' + bK'' = -\frac{2D_3}{\epsilon^2 \underline{\alpha} b^2} + \frac{2D_3}{\epsilon^2 \underline{\alpha} b^2} = 0,
$$

which implies that $N$ is convex and that a critical batch size does not exist. $\qquad \square$

A.5 PROOF OF THEOREM 3.4

Using $K$ defined in Theorem 3.2 leads to the iteration complexity. For example, SGD using **(Decay 4)** satisfies that $K(b) = \frac{1}{\alpha \epsilon^2}(\frac{D_3}{b} + D_1)$, which implies that $\mathcal{K}_\epsilon = O(\frac{1}{\epsilon^2})$. SGD using **(Constant)** satisfies that $N(b) = \frac{C_1 b^2}{\epsilon^2 b - C_2}$ (Theorem 3.3). Using the critical batch size $b^\star = \frac{2C_2}{\epsilon^2}$ in (4) leads to

$$\inf\left\{N : \min_{k \in [0:K-1]} \mathbb{E}[\|\nabla f(\boldsymbol{\theta}_k)\|] \leq \epsilon\right\} \leq N(b^\star) = \frac{4C_1 C_2}{\epsilon^4}, \text{ i.e., } \mathcal{N}_\epsilon = O\left(\frac{1}{\epsilon^4}\right).$$

A similar discussion, together with using $N$ defined in Theorem 3.3 and the critical batch size $b^\star$ in (4), leads to the SFO complexities of **(Decay 1)** and **(Decay 3)**. Using $N$ defined in Theorem 3.3 and a small batch size $b$ leads to the SFO complexities of **(Decay 2)** and **(Decay 4)**. □

A.6 NUMERICAL RESULTS FOR TRAINING DATASETS

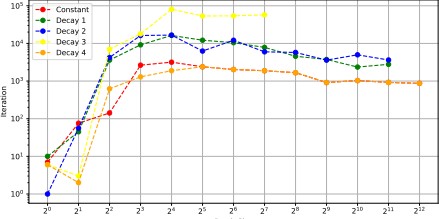

Figure 9: Number of iterations needed for SGD with (Constant), (Decay 1), (Decay 2), (Decay 3), and (Decay 4) to achieve a training accuracy of 0.9 versus batch size (ResNet-18 on CIFAR-10)

Figure 10: SFO complexity needed for SGD with (Constant), (Decay 1), (Decay 2), (Decay 3), and (Decay 4) to achieve a training accuracy of 0.9 versus batch size (ResNet-18 on CIFAR-10)

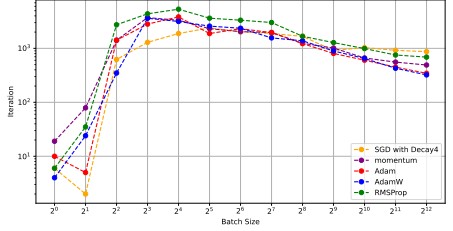
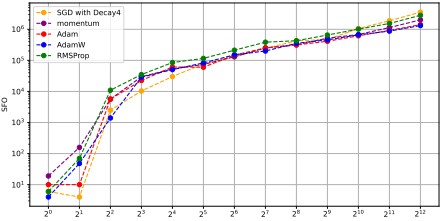

Figure 11: Number of iterations needed for SGD with (Decay 4), momentum, Adam, AdamW, and RMSProp to achieve a training accuracy of 0.9 versus batch size (ResNet-18 on CIFAR-10)

Figure 12: SFO complexity needed for SGD with (Decay 4), momentum, Adam, AdamW, and RM-SProp to achieve a training accuracy of 0.9 versus batch size (ResNet-18 on CIFAR-10)

A.7 TRAINING RESNET-18 ON THE MNIST DATASET

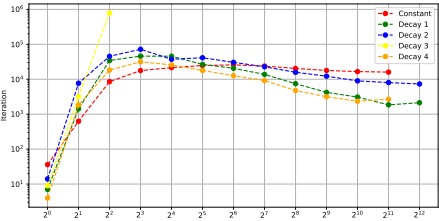

Figure 13: Number of iterations needed for SGD with (Constant), (Decay 1), (Decay 2), (Decay 3), and (Decay 4) to achieve a training accuracy of 0.9 versus batch size (ResNet-18 on CIFAR-100)

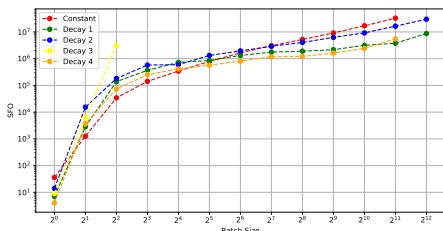

Figure 14: SFO complexity needed for SGD with (Constant), (Decay 1), (Decay 2), (Decay 3), and (Decay 4) to achieve a training accuracy of 0.9 versus batch size (ResNet-18 on CIFAR-100)

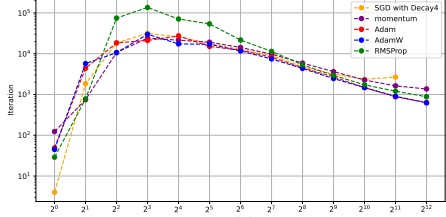

Figure 15: Number of iterations needed for SGD with (Decay 4), momentum, Adam, AdamW, and RMSProp to achieve a training accuracy of 0.9 versus batch size (ResNet-18 on CIFAR-100)

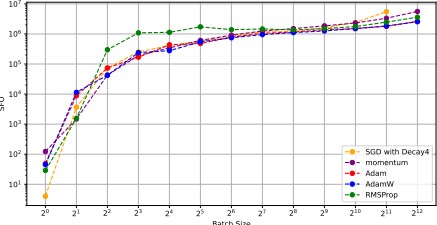

Figure 16: SFO complexity needed for SGD with (Decay 4), momentum, Adam, AdamW, and RM-SProp to achieve a training accuracy of 0.9 versus batch size (ResNet-18 on CIFAR-100)

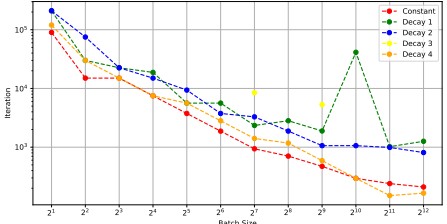

Figure 17: Number of iterations needed for SGD with (Constant), (Decay 1), (Decay 2), (Decay 3), and (Decay 4) to achieve a test accuracy of 0.99 versus batch size (ResNet-18 on MNIST)

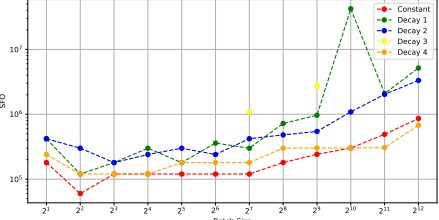

Figure 18: SFO complexity needed for SGD with (Constant), (Decay 1), (Decay 2), (Decay 3), and (Decay 4) to achieve a test accuracy of 0.99 versus batch size (ResNet-18 on MNIST)

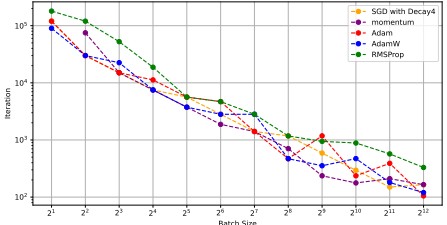

Figure 19: Number of iterations needed for SGD with (Decay 4), momentum, Adam, AdamW, and RMSProp to achieve a test accuracy of 0.99 versus batch size (ResNet-18 on MNIST)

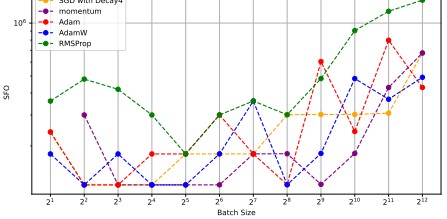

Figure 20: SFO complexity needed for SGD with (Decay 4), momentum, Adam, AdamW, and RM-SProp to achieve a test accuracy of 0.99 versus batch size (ResNet-18 on MNIST)

