# OpenReview forum: "Iteration and Stochastic First-order Oracle Complexities of Stochastic Gradient Descent using Constant and Decaying Learning Rates"
_ICLR.cc/2024/Conference — Submitted to ICLR 2024_

### Official Review · Reviewer_JhF4 · 2023-10-27

**Soundness:** 2 fair
**Presentation:** 2 fair
**Contribution:** 1 poor
**Rating:** 3
**Confidence:** 4

**Summary:**

This paper explores the impact of batch size on the iteration and gradient oracle complexities of the stochastic gradient descent (SGD) algorithm. The objective of the study is to examine how different batch sizes affect the performance of SGD. The paper is written in a reader-friendly manner, making it easily understandable.  In Tables 1 and 2, the authors present a summary of the iteration and gradient oracle complexities of the SGD method using various commonly used step sizes. By presenting this information, the authors offer valuable insights into the behavior of the algorithm. Furthermore, the authors conduct numerical experiments to compare the effectiveness of the step-decay strategy with other optimization algorithms. Through these experiments, they demonstrate the superior performance of step-decay in optimizing the objective function. This finding suggests that step-decay can be a preferable choice when implementing optimization algorithms. However, the contributions of this paper are not sufficient and some of the statements are wrong.

**Strengths:**

This paper provides a thorough investigation into the relationship between batch size and the complexities of the SGD algorithm. The authors present their findings in a clear and concise manner, making them accessible to readers.

**Weaknesses:**

The paper is not well ready yet and contributions are trivial. As I checked, the analysis of SGD is quite simple and there is no technical challenge in the analysis. Besides, the main statements on step-decay are wrong. Please see the reasons below.

To calculate the iteration and gradient oracle complexity of the step-decay method, it is crucial to consider the impact of the lower bound values, denoted as $\underline{\alpha}$ and $T$ (representing the length of each stage for step-decay). Unfortunately, the authors of the paper have overlooked this important aspect, which is an incorrect approach. The reason why considering the lower bound values is essential lies in the relationship between $\underline{\alpha}$, $T$, and the total number of iterations, denoted as $K$. Specifically, we have the inequality $\underline{\alpha} \leq \alpha \eta^{p-1}$, where $p = K/T$. This inequality implies that the lower bound value $\underline{\alpha}$ should be taken into account when determining the iteration and gradient oracle complexities of the step-decay method. Ignoring this relationship can lead to flawed conclusions and inaccurate assessments of the algorithm's performance. Therefore, the related complexities results on step-decay are wrong.

Other weaknesses or typos:
1. In the abstract, the authors made a claim that "SGD using a step-decay learning rate and a small batch size reduces the SFO (Stochastic First-Order) complexity to find a local minimizer of a loss function." However, upon reviewing the paper, it becomes apparent that the study primarily focuses on demonstrating the convergence of SGD to a stationary point rather than specifically proving convergence to a local minimizer.

**Questions:**

See the weakness above

---

> ### Author Response · Authors · 2023-11-19
> **Reply to Reviewer JhF4's comments**
>
> **Comment 1:**
> The paper is not well ready yet and contributions are trivial. As I checked, the analysis of SGD is quite simple and there is no technical challenge in the analysis. Besides, the main statements on step-decay are wrong. Please see the reasons below.
>
> To calculate the iteration and gradient oracle complexity of the step-decay method, it is crucial to consider the impact of the lower bound values, denoted as $\underline{\alpha}$ and $T$ (representing the length of each stage for step-decay). Unfortunately, the authors of the paper have overlooked this important aspect, which is an incorrect approach. The reason why considering the lower bound values is essential lies in the relationship between $\underline{\alpha}$, $T$, and the total number of iterations, denoted as $K$. Specifically, we have the inequality $\underline{\alpha} \leq \alpha\eta^{p-1}$, where $p = K/T$. This inequality implies that the lower bound value $\underline{\alpha}$ should be taken into account when determining the iteration and gradient oracle complexities of the step-decay method. Ignoring this relationship can lead to flawed conclusions and inaccurate assessments of the algorithm's performance. Therefore, the related complexities results on step-decay are wrong.
>
> **Reply:**
> We apologize for the insufficient analysis of Decay 4. Based on [1], we would like to revise our analysis of Decay 4.
>
> First, we modify the definition of Decay 4 as follows. Let $\alpha_0 > 0$, $\beta \geq 2$, $T, P \geq 1$, and $K = TP$.
> A step-decay learning rate is
> $$(\alpha_k)_{k=0}^{K-1}
> = \left(\underbrace{\alpha_0, \alpha_0, \cdots, \alpha_0}_T,
> \frac{\alpha_0}{\beta}, \frac{\alpha_0}{\beta}, \cdots, \frac{\alpha_0}{\beta}, \cdots,
> \frac{\alpha_0}{\beta^{P-1}}, \frac{\alpha_0}{\beta^{P-1}}, \cdots, \frac{\alpha_0}{\beta^{P-1}} \right),$$
> which is monotone decreasing for $k$.
> We assume that $\alpha_0 \leq \frac{1}{L}$, which implies that, for all $k \in [0:K-1]$, $\alpha_k \leq \frac{1}{L}$.
>
> Next, we can show that
> \begin{align*}
> \alpha_k \left(1 - \frac{L \alpha_k}{2} \right) \mathbb{E} \left[ \Vert  \nabla f({\theta}_{k}) \Vert^2 \right]
> \leq \mathbb{E} [ f (\theta_k)-f (\theta_k+1)]+\frac{L \sigma^2 \alpha_k^2}{2b}
> \end{align*}
> based on Lemma A.1 (our paper).
>
> We can thus modify the result of Decay 4 in Theorem 3.1 as follows:
>
> Let $P = \log_{\alpha_0} K/2$. Then, the sequence $(\theta_k)$ generated by Algorithm 1 under (C1)-(C3) satisfies that, for all $K \geq 1$,
> \begin{align*}
> \min_{k \in [0:K-1]} \mathbb{E}\left[\Vert \nabla f({\theta}_k) \Vert^2 \right]
> \leq
> \frac{D_3 (\sqrt{K}-2 )}{\sqrt{K}-1}+\frac{D_4}{(\sqrt{K}-1)b},
> \end{align*}
> where $D_3 = \frac{(\beta -1) \Delta}{\alpha_0}$ and $D_4 = \alpha_0 L \sigma^2 (\beta -1)$.
>
> Then, we have that $K$ and $N$ of SGD using Decay 4 needed to an $\epsilon$-approximation are
> $$K(b) = \frac{1}{(D_3 - \epsilon^2)^2} \left( (2D_3 - \epsilon^2)^2  - \frac{2 D_4 (2 D_3 -  \epsilon^2)}{b} + \frac{D_4^2}{b^2} \right).$$
> $$N(b) = \frac{b}{(D_3 - \epsilon^2)^2} \left( (2D_3 - \epsilon^2)^2  - \frac{2 D_4 (2 D_3 -  \epsilon^2)}{b} + \frac{D_4^2}{b^2} \right).$$
> As a result, we have the following:
> - The iterations $K(b)$ is monotone decreasing for $b < \frac{D_4}{2D_3 - \epsilon^2}$ and convex for $b < \frac{3 D_4}{2(2 D_3 - \epsilon^2)}$ .
> - The SFO complexity $N(b)$ is convex for $b > 0$ and $N'(b) > 0$ holds for all $b > 0$.
> - We have that $K_\epsilon = O(1/\epsilon^4)$ and $N_\epsilon = O(1/\epsilon^4)$.
>
> We will revise the manuscript based on the above discussion.
>
> [1] Wang, Xiaoyu, Sindri Magnússon, and Mikael Johansson. "On the convergence of step decay step-size for stochastic optimization." Advances in Neural Information Processing Systems 34 (2021): 14226-14238.
>
> **Comment 2:**
> Other weaknesses or typos:
> 1. In the abstract, the authors made a claim that "SGD using a step-decay learning rate and a small batch size reduces the SFO (Stochastic First-Order) complexity to find a local minimizer of a loss function." However, upon reviewing the paper, it becomes apparent that the study primarily focuses on demonstrating the convergence of SGD to a stationary point rather than specifically proving convergence to a local minimizer.
>
> **Reply:**
> Thank you for pointing out. I will revise our claim as "SGD using a step-decay learning rate and a small batch size reduces the SFO complexity to find a stationary point of a loss function."

---

> > ### Comment · Reviewer_JhF4 · 2023-11-22
> > **Response to Authors**
> >
> > Thank you for your detailed responses and changes made in the rebuttal. However, this paper is not in good shape yet and I will keep my score.

---

### Official Review · Reviewer_1sx5 · 2023-10-29

**Soundness:** 1 poor
**Presentation:** 3 good
**Contribution:** 1 poor
**Rating:** 3
**Confidence:** 4

**Summary:**

This paper studies the stochastic first-order oracle complexity (SFO, defined by this paper) of SGD with diminishing and constant learning rates. It shows that SGD using a step-decay learning rate and a small batch size achieves the best performance in terms of SFO complexity. Numerical experiments are provided.

**Strengths:**

1. The paper is well-written and I enjoy reading it.

2. Diminishing learning rate is commonly adopted in deep learning, and it is thus important to study it.

**Weaknesses:**

1. The stochastic gradient is generated in a different way from practice, and I think this should be highlighted. Specifically, in each iteration, this paper assumes the stochastic gradient is chosen as an ensemble as individual gradients sampled with replacement from some distribution. However, in deep learning practice, the individual gradient is sampled without-replacement and this leads to a gap between practice and the presented theory.

2. I do not think SFO is a reasonable measure. In practice, different individual gradients are calculated parallelly and the corresponding time does not accumulate across samples.

3. The definition of K_{\epsilon} and N_{epsilon} seems to be weird, since the learning rate does not appear in any side of the equation.

4. I wonder what is the novelty of Theorem 3.1. Is not it a very basic analysis of SGD?

5. I find the result of Decay 4 problematic. Specifically, in Theorem 3.2, isn't $\underline{\alpha}$ itself depends on $K(b)$? How can $K(b)$ be further calculated by $\underline{\alpha}$? That being said, when $T$ is independent of $\epsilon$, T is in the same order as $\varepsilon$ as P. Therefore, $\underline{\alpha}$ depends exponentially on $K$. Applying this to Theorem 3.2, it indicates $K(b)$ is also exponentially dependent over $\varepsilon$ and contradicts Theorem 3.4.

**Questions:**

1. On page 3, Is $N_{\epsilon}$ just $bK_{\epsilon}$?If yes, why not use the simpler one?

---

> ### Author Response · Authors · 2023-11-18
> **Reply to Reviewer 1sx5's comments**
>
> **Comment 1:**
> The stochastic gradient is generated in a different way from practice, and I think this should be highlighted. Specifically, in each iteration, this paper assumes the stochastic gradient is chosen as an ensemble as individual gradients sampled with replacement from some distribution. However, in deep learning practice, the individual gradient is sampled without-replacement and this leads to a gap between practice and the presented theory.
>
> **Reply:**
> Thank you for your valuable comment. Our paper assumes that, at iteration $k$, $b$ gradients of the loss functions are sampled randomly and the stochastic gradient is defined by the mean of $b$ gradients. We believe that the first-order oracle used in our paper is natural.
>
> **Comment 2:**
> I do not think SFO is a reasonable measure. In practice, different individual gradients are calculated parallelly and the corresponding time does not accumulate across samples.
>
> **Reply:**
> Thank you for your valuable comment. It is important to evaluate the implementation time of SGD. In this paper, in both theory and practice, we study the relationship between the batch size and the iteration complexity/the SFO complexity without computing $b$ gradients parallelly.
>
> **Comment 3:**
> The definition of $K_{\epsilon}$ and $N_{\epsilon}$ seems to be weird, since the learning rate does not appear in any side of the equation.
>
> **Reply:**
> Thank you again for your comment. In the revision, we would like to replace (2) with $K_\epsilon (n,b,\alpha_k,\Delta,L,\sigma^2)$ and  $N_\epsilon (n,b,\alpha_k,\Delta,L,\sigma^2)$.
>
> **Comment 4:**
> I wonder what is the novelty of Theorem 3.1. Is not it a very basic analysis of SGD?
>
> **Reply:**
> Theorem 3.1 is used to provide Theorems 3.2 and 3.3. Our main results are Theorems 3.2 and 3.3. Hence, we may replace Theorem 3.1 by e.g., Lemma 3.1/Proposition 3.1.
>
> **Comment 5:**
> I find the result of Decay 4 problematic. Specifically, in Theorem 3.2, isn’t $\underline{\alpha}$ itself depends on $K(b)$? How can $K(b)$ be further calculated by $\underline{\alpha}$? That being said, when $T$ is independent of $\epsilon$, $T$ is in the same order as $\epsilon$ as $P$. Therefore, $\underline{\alpha}$ depends exponentially on $K$. Applying this to Theorem 3.2, it indicates $K(b)$ is also exponentially dependent over $\epsilon$ and contradicts Theorem 3.4.
>
> **Reply:**
> We apologize for the insufficient analysis of Decay 4. Based on [1], we would like to revise our analysis of Decay 4.
>
> First, we modify the definition of Decay 4 as follows. Let $\alpha_0 > 0$, $\beta \geq 2$, $T, P \geq 1$, and $K = TP$.
> A step-decay learning rate is
> $$(\alpha_k)_{k=0}^{K-1}
> = \left(\underbrace{\alpha_0, \alpha_0, \cdots, \alpha_0}_T,
> \frac{\alpha_0}{\beta}, \frac{\alpha_0}{\beta}, \cdots, \frac{\alpha_0}{\beta}, \cdots,
> \frac{\alpha_0}{\beta^{P-1}}, \frac{\alpha_0}{\beta^{P-1}}, \cdots, \frac{\alpha_0}{\beta^{P-1}} \right),$$
> which is monotone decreasing for $k$.
> We assume that $\alpha_0 \leq \frac{1}{L}$, which implies that, for all $k \in [0:K-1]$, $\alpha_k \leq \frac{1}{L}$.
>
> Next, we can show that
> \begin{align*}
> \alpha_k \left(1 - \frac{L \alpha_k}{2} \right) \mathbb{E} \left[ \Vert  \nabla f({\theta}_{k}) \Vert^2 \right]
> \leq \mathbb{E} [ f (\theta_k)-f (\theta_k+1)]+\frac{L \sigma^2 \alpha_k^2}{2b}
> \end{align*}
> based on Lemma A.1 (our paper).
>
> We can thus modify the result of Decay 4 in Theorem 3.1 as follows:
>
> Let $P = \log_{\alpha_0} K/2$. Then, the sequence $(\theta_k)$ generated by Algorithm 1 under (C1)-(C3) satisfies that, for all $K \geq 1$,
> \begin{align*}
> \min_{k \in [0:K-1]} \mathbb{E}\left[\Vert \nabla f({\theta}_k) \Vert^2 \right]
> \leq
> \frac{D_3 (\sqrt{K}-2 )}{\sqrt{K}-1}+\frac{D_4}{(\sqrt{K}-1)b},
> \end{align*}
> where $D_3 = \frac{(\beta -1) \Delta}{\alpha_0}$ and $D_4 = \alpha_0 L \sigma^2 (\beta -1)$.
>
> Then, we have that $K$ and $N$ of SGD using Decay 4 needed to an $\epsilon$-approximation are
> $$K(b) = \frac{1}{(D_3 - \epsilon^2)^2} \left( (2D_3 - \epsilon^2)^2  - \frac{2 D_4 (2 D_3 -  \epsilon^2)}{b} + \frac{D_4^2}{b^2} \right).$$
> $$N(b) = \frac{b}{(D_3 - \epsilon^2)^2} \left( (2D_3 - \epsilon^2)^2  - \frac{2 D_4 (2 D_3 -  \epsilon^2)}{b} + \frac{D_4^2}{b^2} \right).$$
> As a result, we have the following:
> - The iterations $K(b)$ is monotone decreasing for $b < \frac{D_4}{2D_3 - \epsilon^2}$ and convex for $b < \frac{3 D_4}{2(2 D_3 - \epsilon^2)}$ .
> - The SFO complexity $N(b)$ is convex for $b > 0$ and $N'(b) > 0$ holds for all $b > 0$.
> - We have that $K_\epsilon = O(1/\epsilon^4)$ and $N_\epsilon = O(1/\epsilon^4)$.
>
> We will revise the manuscript based on the above discussion.
>
> [1] Wang, Xiaoyu, Sindri Magnússon, and Mikael Johansson. “On the convergence of step decay step-size for stochastic optimization.” Advances in Neural Information Processing Systems 34 (2021): 14226-14238.
>
> **Comment 6:**
> On page 3, Is $N_\epsilon$ just $bK_\epsilon$?If yes, why not use the simpler one?
>
> **Reply:**
> Thank you. We will revise it accordingly.

---

> ### Comment · Reviewer_1sx5 · 2023-11-22
> **Post-rebuttal**
>
> I thank the authors for the detailed response. The authors promised to throughout revise the results regarding Decay 4, which is a core result in this paper and thus I think another round of review is needed for these new results. I will keep my score.

---

### Official Review · Reviewer_BdrQ · 2023-11-01

**Soundness:** 2 fair
**Presentation:** 2 fair
**Contribution:** 3 good
**Rating:** 3
**Confidence:** 4

**Summary:**

This paper studies the complexity of computing stationary points with SGD
using a variety of step-size schedules. The authors derive convergence rates
with an explicit dependence on the batch-size for SGD with a constant step-size
as well as polynomial decay and step-decay step-size schedules. These rates
are then converted into iteration and oracle complexities and studied as a
function of the mini-batch size.
The authors prove that the parameterized complexities are convex functions
and use this to derive optimal batch-sizes for different schedules.
The submission concludes with experiments comparing different schedules on
CIFAR-10 and CIFAR-100.

**Strengths:**

The main strength of this paper is its novel approach to hyperparameter
tuning for SGD. While it is typical to tune (at least in theory) the step-size
parameter to minimize the oracle complexity, maintaining an explicit dependence
on the mini-batch size in the convergence rate and using this to understand
the trade-offs between iteration and oracle complexity is an interesting idea.
In addition to this, the paper has the following strengths:

- The authors provide a simple and clear analysis for SGD which covers
    SGD with a fixed step-size, polynomial decay, and step-decay
    schedules.

- Although the optimal batch-sizes for fixed step-sizes and polynomial decay schedules
    depend on unknown parameters of the problem, understanding the optimal values
    may allow for new heuristics for selecting the batch-size in practice.

- The experiments, although simple, generally reflect the theory and show that
    tuning the batch-size can lead to improvements in optimization given a fixed
    budget of gradient evaluations.

**Weaknesses:**

This paper has several significant weaknesses that should be addressed before
publication. In particular,

- The convergence rate given for SGD with the step-decay schedule is misleading
    and leads to incorrect iteration and oracle complexities for this method.

- The paper does not address the fact that $b \leq n$ must be maintained, where
    $n$ is the number of functions in the finite sum. As a result, the optimal
    batch-sizes which the authors derive may not be attainable depending on the
    desired precision of the solution. For example, as $\epsilon \rightarrow 0$,
    $b \rightarrow \infty$ for a fixed step-size.

- The manuscript is unnecessarily "mathy" and many equations could be
    omitted while maintaining the same results. See, for example, Equations 1 and 2.
    While this makes the writing seem superficially impressive, it is difficult to
    read and detracts from the flow of the paper.

- None of the experiments serve to verify the theoretical derivations.
    I would liked to see at least one synthetic experiment for which the problem
    constants are known (e.g. a simple quadratic) and $b^*$ can be computed.
    Plots similar to that in Figures 1/2 could then show that $b^*$ does, in fact,
    obtain the optimal oracle complexity as claimed.

Given these issues, I cannot recommend accepting the submission at this time.
However, I am willing to increase my score if they address these issues. At
a very minimum, I feel the problem with the complexity of the step-size schedule
must be resolved. See "Questions" below for more details.

**Questions:**

- "SGD using a decaying learning rate...": some additional comment on the type
    of learning rate decay is needed here. SGD with step-size $\alpha_k = 1 / \log(k)$
    does not converge as $O(1/\sqrt{K})$ despite $\alpha_k \rightarrow 0$.

- First math display in Section 1.3.2: this bound should mentioned somewhere that
    $b > n$ isn't possible and $b = n$ reduces to full-batch gradient descent.
    As a result, it is not always feasible to select the batch-size to minimize
    the oracle complexity.

- "Accordingly, small batch sizes are appropriate for a decaying learning rate or a
    step-decay learning rate": why is this true? You have said that SFO complexity
    has no positive stationary point, but that doesn't imply it is increasing
    in $b$ or that a small batch-size minimize the complexity over the positive
    integers. Can you please address this fact?

- Equation (2) and Table 2: It is somewhat confusing to switch from measuring
    convergence using the squared gradient norm in Table 1 to convergence of
    just the gradient norm in Table 2 and Equation (2).

- Theorem 3.1: I am concerned by the presentation of the convergence rates for
    SGD with step-decay. Firstly, $\underline{\alpha}$ does not appear to be
    defined anywhere. From the proof in the appendix, it seems
    $\underline{\alpha} = \alpha_{K-1} = \alpha \eta^{K/T-1}$.  This quantity
    depends on $K$ --- it is exponentially decreasing every $K/T$ iterations
    --- so that it is incorrect to write it as a constant factor.  Similarly,
    $D_3$ depends on $T$, which may or may not have a relationship with $K$
    depending on algorithm parameters.

    Only be carefully optimizing over $T$ can a final rate of convergence be
    obtained. Wang et al. [1] set $T = K / \log_{\eta}(K)$ to obtain a final
    convergence rate of $O(\log(T)/\sqrt{T})$.  In contrast, treating
    $\underline{\alpha}$ as a constant leads to an deceptive presentation of
    the convergence rate in Table 1. Moreover, I am fairly certain the
    complexity of $O(1/\epsilon^2)$ for computing an $\epsilon$ stationary
    point in Table 2 is incorrect and violates lower bounds due to Drori and
    Shamir [2].

- Theorem 3.4: In addition to the issue with the complexity of step-decay raised
     previously, this theorem assumes that $b \leq n$ can be chosen arbitrarily
     large in order to obtain the desired complexity. For example, SGD with a
     constant step-size requires $b \geq 2 C_2 \epsilon$, which diverges to
     infinity as $\epsilon \rightarrow 0$. But this is not sensible because
     $n$ is assumed to be a fixed, finite number of training examples.
     If this is not the cause, then the authors must specify somewhere that they
     assume a setting where $n$ can be taken arbitrarily large.

### References

[1] Wang, Xiaoyu, Sindri Magnússon, and Mikael Johansson. "On the convergence
of step decay step-size for stochastic optimization." Advances in Neural
Information Processing Systems 34 (2021): 14226-14238.

[2] Drori, Yoel, and Ohad Shamir. "The complexity of finding stationary points
with stochastic gradient descent." International Conference on Machine
Learning. PMLR, 2020.

---

> ### Author Response · Authors · 2023-11-19
> **Reply to Reviewer BdrQ's comments**
>
> **Comment 1:**
> The convergence rate given for SGD with the step-decay schedule is misleading and leads to incorrect iteration and oracle complexities for this method.
>
> **Comment 9:**
> Theorem 3.1: I am concerned by the presentation of the convergence rates for SGD with step-decay. Firstly, $\underline{\alpha}$ does not appear to be defined anywhere. From the proof in the appendix, it seems $\underline{\alpha}=\alpha_{K-1}=\alpha\eta^{K/T-1}$. This quantity depends on $K$ --- it is exponentially decreasing every $K/T$ iterations --- so that it is incorrect to write it as a constant factor. Similarly, $D_3$ depends on $T$, which may or may not have a relationship with $K$ depending on algorithm parameters. \\
> Only be carefully optimizing over T can a final rate of convergence be obtained. Wang et al. [1] set $T=\frac{K}{\log_{\eta}(K)}$ to obtain a final convergence rate of $O(\frac{\log(T)}{\sqrt{T}})$. In contrast, treating $\underline{\alpha}$ as a constant leads to an deceptive presentation of the convergence rate in Table 1. Moreover, I am fairly certain the complexity of $O(\frac{1}{\epsilon^2})$ for computing an $\epsilon$ stationary point in Table 2 is incorrect and violates lower bounds due to Drori and Shamir [2].
>
> **Reply:**
> We apologize for the insufficient analysis of Decay 4. Based on [1], we would like to revise our analysis of Decay 4.
>
> First, we modify the definition of Decay 4 as follows. Let $\alpha_0 > 0$, $\beta \geq 2$, $T, P \geq 1$, and $K = TP$.
> A step-decay learning rate is
> $$(\alpha_k)_{k=0}^{K-1}
> = \left(\underbrace{\alpha_0, \alpha_0, \cdots, \alpha_0}_T,
> \frac{\alpha_0}{\beta}, \frac{\alpha_0}{\beta}, \cdots, \frac{\alpha_0}{\beta}, \cdots,
> \frac{\alpha_0}{\beta^{P-1}}, \frac{\alpha_0}{\beta^{P-1}}, \cdots, \frac{\alpha_0}{\beta^{P-1}} \right),$$
> which is monotone decreasing for $k$.
> We assume that $\alpha_0 \leq \frac{1}{L}$, which implies that, for all $k \in [0:K-1]$, $\alpha_k \leq \frac{1}{L}$.
>
> Next, we can show that
> \begin{align*}
> \alpha_k \left(1 - \frac{L \alpha_k}{2} \right) \mathbb{E} \left[ \Vert  \nabla f({\theta}_{k}) \Vert^2 \right]
> \leq \mathbb{E} [ f (\theta_k)-f (\theta_k+1)]+\frac{L \sigma^2 \alpha_k^2}{2b}
> \end{align*}
> based on Lemma A.1 (our paper).
>
> We can thus modify the result of Decay 4 in Theorem 3.1 as follows:
>
> Let $P = \log_{\alpha_0} K/2$. Then, the sequence $(\theta_k)$ generated by Algorithm 1 under (C1)-(C3) satisfies that, for all $K \geq 1$,
> \begin{align*}
> \min_{k \in [0:K-1]} \mathbb{E}\left[\Vert \nabla f({\theta}_k) \Vert^2 \right]
> \leq
> \frac{D_3 (\sqrt{K}-2 )}{\sqrt{K}-1}+\frac{D_4}{(\sqrt{K}-1)b},
> \end{align*}
> where $D_3 = \frac{(\beta -1) \Delta}{\alpha_0}$ and $D_4 = \alpha_0 L \sigma^2 (\beta -1)$.
>
> Then, we have that $K$ and $N$ of SGD using Decay 4 needed to an $\epsilon$-approximation are
> $$K(b) = \frac{1}{(D_3 - \epsilon^2)^2} \left( (2D_3 - \epsilon^2)^2  - \frac{2 D_4 (2 D_3 -  \epsilon^2)}{b} + \frac{D_4^2}{b^2} \right).$$
> $$N(b) = \frac{b}{(D_3 - \epsilon^2)^2} \left( (2D_3 - \epsilon^2)^2  - \frac{2 D_4 (2 D_3 -  \epsilon^2)}{b} + \frac{D_4^2}{b^2} \right).$$
> As a result, we have the following:
> - The iterations $K(b)$ is monotone decreasing for $b < \frac{D_4}{2D_3 - \epsilon^2}$ and convex for $b < \frac{3 D_4}{2(2 D_3 - \epsilon^2)}$ .
> - The SFO complexity $N(b)$ is convex for $b > 0$ and $N'(b) > 0$ holds for all $b > 0$.
> - We have that $K_\epsilon = O(1/\epsilon^4)$ and $N_\epsilon = O(1/\epsilon^4)$.
>
> We will revise the manuscript based on the above discussion.
>
> **Comment 2:**
> The paper does not address the fact that $b \leq n$ must be maintained, where $n$ is the number of functions in the finite sum. As a result, the optimal batch-sizes which the authors derive may not be attainable depending on the desired precision of the solution. For example, as $\epsilon \to 0$, $b \to \infty$ for a fixed step-size.
>
> **Comment 6:**
> First math display in Section 1.3.2: this bound should mentioned somewhere that $b>n$ isn't possible and $b=n$ reduces to full-batch gradient descent. As a result, it is not always feasible to select the batch-size to minimize the oracle complexity.
>
> **Reply:**
> Thank you for pointing out. We will add the condition $b \leq n$ to (C3). Also, we will revise Section 1.3.2 accordingly.
>
> **Comment 3:**
> The manuscript is unnecessarily "mathy" and many equations could be omitted while maintaining the same results. See, for example, Equations 1 and 2. While this makes the writing seem superficially impressive, it is difficult to read and detracts from the flow of the paper.
>
> **Reply:**
> We kindly agree with your comment. We will revise the manuscript accordingly.

---

> > ### Author Response · Authors · 2023-11-19
> > **Reply to Reviewer BdrQ's comments**
> >
> > **Comment 4:**
> > None of the experiments serve to verify the theoretical derivations. I would liked to see at least one synthetic experiment for which the problem constants are known (e.g. a simple quadratic) and $b^*$ can be computed. Plots similar to that in Figures 1/2 could then show that $b^*$ does, in fact, obtain the optimal oracle complexity as claimed.
> >
> > **Comment 10:**
> > Theorem 3.4: In addition to the issue with the complexity of step-decay raised previously, this theorem assumes that $b\leq n$ can be chosen arbitrarily large in order to obtain the desired complexity. For example, SGD with a constant step-size requires $b \geq 2C_{2\epsilon}$, which diverges to infinity as $\epsilon \to 0$. But this is not sensible because $n$ is assumed to be a fixed, finite number of training examples. If this is not the cause, then the authors must specify somewhere that they assume a setting where $n$ can be taken arbitrarily large.
> >
> > **Reply:**
> > Thank you for your valuable comments. In the revision, we clarify that $n$ can be taken arbitrarily large and $\epsilon$ is a sufficiently small positive constant.
> >
> > We now prepare a synthetic experiment. Hence, we plan to reply further comments to Comment 4 as soon as possible.
> >
> > **Comment 5:**
> > "SGD using a decaying learning rate...": some additional comment on the type of learning rate decay is needed here. SGD with step-size $\alpha_k = \frac{1}{\log\left(k\right)}$ does not converge as $\mathcal{O}\left(\frac{1}{\sqrt{K}}\right)$ despite $\alpha_k \to 0$.
> >
> > **Reply:**
> > Thank you for your valuable comment. In the revision, we would like to discuss SGD with $\alpha_k = \frac{1}{\log k}$.
> >
> > **Comment 7:**
> > "Accordingly, small batch sizes are appropriate for a decaying learning rate or a step-decay learning rate": why is this true? You have said that SFO complexity has no positive stationary point, but that doesn't imply it is increasing in b or that a small batch-size minimize the complexity over the positive integers. Can you please address this fact?
> >
> > **Reply:**
> > Thank you for your valuable comment. In (Decay 2), $N(b)$ is convex for $b > 0$ (see Theorem 3.3). Moreover, in (Decay 2), $b^\star < 0$ holds, and hence, $N(b)$ is monotonically increasing for $b>0$. In (Decay 4), we have that $N(b)$ is convex for $b > 0$ and $N'(b) > 0$ holds for all $b > 0$. Therefore, small batch sizes are appropriate for (Decay 2) and (Decay 4) in the sense of minimizing the SFO complexities.
> >
> > **Comment 8:**
> > Equation (2) and Table 2: It is somewhat confusing to switch from measuring convergence using the squared gradient norm in Table 1 to convergence of just the gradient norm in Table 2 and Equation (2).
> >
> > **Reply:**
> > We kindly agree with your comment. In the revision, we will unify the performance measure as the gradient norm.

---

> > > ### Comment · Reviewer_BdrQ · 2023-11-20
> > >
> > > I would like to thank the authors for responding to my review.
> > >
> > > While I appreciate the proposed modifications to the analysis of step-decay, I think such a significant change may deserve another round of reviews.  The issues raised by Reviewer xRbi, i.e. existing mini-batch analyses for SGD with constant step-size, are also significant and reduce the novelty of this work. Considering both of these issues, I will maintain my score for now.

---

### Official Review · Reviewer_xRbi · 2023-11-09

**Soundness:** 2 fair
**Presentation:** 2 fair
**Contribution:** 1 poor
**Rating:** 1
**Confidence:** 5

**Summary:**

This manuscript studied the effects of batch size and learning rate for nonconvex smooth optimization. The authors established iteration complexity and SFO (Stochastic First Order Oracle) complexity of the problem.

Despite the result is interesting, most of the results are known (or straightforward extension) in the literature.

**Strengths:**

The paper is well-written.

**Weaknesses:**

The paper's theoretical results are known in the literature (e.g., [r1], [r2]). The hardness result [r3] says that: whatever batch size the SGD algorithm can choose, the SFO cannot be better than $O(1/\epsilon^4)$.

[r1] Ghadimi, Saeed, and Guanghui Lan. "Stochastic first-and zeroth-order methods for nonconvex stochastic programming." SIAM Journal on Optimization 23, no. 4 (2013): 2341-2368.

[r2] Ghadimi, S., Lan, G., & Zhang, H. (2016). Mini-batch stochastic approximation methods for nonconvex stochastic composite optimization. Mathematical Programming, 155(1-2), 267-305.

[r3] Arjevani, Yossi, Yair Carmon, John C. Duchi, Dylan J. Foster, Nathan Srebro, and Blake Woodworth. "Lower bounds for non-convex stochastic optimization." Mathematical Programming 199, no. 1-2 (2023): 165-214.

**Questions:**

Can you describe how your approach is better than the references I gave above (e.g., [r1, r2])?

---

> ### Author Response · Authors · 2023-11-18
> **Reply to Reviewer xRbi's comments**
>
> **Comment:**
> The paper's theoretical results are known in the literature (e.g., [r1], [r2]). The hardness result [r3] says that: whatever batch size the SGD algorithm can choose, the SFO cannot be better than $O(1/\epsilon^4)$.
>
> [r1] Ghadimi, Saeed, and Guanghui Lan. "Stochastic first-and zeroth-order methods for nonconvex stochastic programming." SIAM Journal on Optimization 23, no. 4 (2013): 2341-2368.
>
> [r2] Ghadimi, S., Lan, G., Zhang, H. (2016). Mini-batch stochastic approximation methods for nonconvex stochastic composite optimization. Mathematical Programming, 155(1-2), 267-305.
>
> [r3] Arjevani, Yossi, Yair Carmon, John C. Duchi, Dylan J. Foster, Nathan Srebro, and Blake Woodworth. "Lower bounds for non-convex stochastic optimization." Mathematical Programming 199, no. 1-2 (2023): 165-214.
>
> **Questions:**
> Can you describe how your approach is better than the references I gave above (e.g., [r1, r2])?
>
> **Reply:**
> Thank you for your valuable comments.
>
> Assumptions (C1)--(C3) (our paper) are general and well used in previous papers. From the experimental results in the previous paper [1] and our experimental results (see Figures 2 and 6, and so on), it is clear that the SFO complexity of SGD depends on both the learning rate and the batch size. Therefore, we study the relationship between batch size and the SFO complexity of SGD using specific learning rates (Constant and Decay 1--4) from the viewpoints of both theory and practice. Our results show that
>
> - The number of iterations $K$ needed for nonconvex optimization of SGD is a monotone decreasing and convex function with respect to the batch size $b$ (Theorem 3.2);
> - The SFO complexity $N$ is convex with respect to the batch size $b$ (Theorem 3.3);
> - The existence of the critical batch size that is a global minimizer of $N$ depends on learning rates (Theorem 3.3).
>
> The above results (Theorems 3.2 and 3.3) are our contributions compared with the previous results in [r1] and [r2].
>
> Reference [r3] showed that the iteration complexity of the general SGD does not perform better than the order of $\frac{1}{\epsilon^4}$. However, we would like to emphasize that there is a room such that, if specific learning rate and batch size are used, then SGD will perform better than the order of $\frac{1}{\epsilon^4}$. Our paper shows that, if we can use a constant learning rate $\alpha < 2/L$ and the critical batch size $b^* = 2 C_2/\epsilon^2$, then the iteration complexity breaks through the order of $\frac{1}{\epsilon^4}$ and becomes $O(\frac{1}{\epsilon^2})$. We can also check that using specific batch size performs well in practice. For example, Figure 2 (our paper) indicates that the SFO complexity of SGD using $b = 2^4$ is better than the ones using other batch sizes.
>
> [1] Christopher J. Shallue, Jaehoon Lee, Joseph Antognini, Jascha Sohl-Dickstein, Roy Frostig, and
> George E. Dahl. Measuring the effects of data parallelism on neural network training. Journal of
> Machine Learning Research, 20:1–49, 2019.

---

> ### Comment · Reviewer_xRbi · 2023-11-18
> **Response**
>
> I would like thank the authors' response. However it does not change my mind.
>
> 1. If your batch size is $O(1/\epsilon^2)$ and your iteration complexity is $O(1/\epsilon^2)$ then the total gradient complexity is still $O(1/\epsilon^4)$.
>
> 2. The authors claim that "To the best of our knowledge, this is the first paper to provide that the iteration complexity of SGD using a constant learning rate and the critical batch size is $O(\frac{1}{\epsilon^2})". This statement is very wrong. Could you please check reference [r2] carefully before claiming anything?
>
> I will end my discussion here.

---

> > ### Author Response · Authors · 2023-11-19
> > **Reply to Reviewer xRbi's comments**
> >
> > We sincerely apologize for our claim without checking carefully the previous results.
> >
> > Corollaries 6 and 7 in [r2] are the same as our results for SGD with a constant learning rate in Theorems 3.1 and 3.3, since the randomized stochastic projected gradient free algorithm in [r2] that is one of stochastic zeroth-order (SZO) methods and coincides with SGD can be applied to the situations where only noisy function values are available. In particular, Corollary 6 shows the convergence rate of the SZO method using a fixed batch size and Corollary 7 indicates the SZO complexity that is the same as the SFO complexity (our paper).  Hence, Corollaries 6 and 7 in [r2] lead to the finding that the iteration complexity of SGD is $O(1/\epsilon^2)$ and the SFO complexity of SGD is $O(1/\epsilon^4)$.
> >
> > Therefore, we deleted our claim such that “To the best of our knowledge, this is the first paper to provide that the iteration complexity of SGD using a constant learning rate and the critical batch size is $O(1/\epsilon^2)$.”

---

### Meta-Review · Area_Chair_bR5c · 2023-12-06

**Metareview:**

This paper studies the relationship between the batch size and the iteration/stochastic first-order oracle complexities of the stochastic gradient descent (SGD) method. Although the reviewers find the paper well written and the problem studied important, they also raise several serious concerns: (i) There is insufficient comparison with results in existing literature. (ii) The novelty of the contributions is limited. (iii) The convergence analysis of the SGD with step-decay is not convincing and requires substantial elaboration. Based on the above, I regrettably have to recommend rejection of the manuscript.

**Justification For Why Not Higher Score:**

The contributions in the current manuscript lack novelty. Moreover, the reviewers express doubts on the correctness of the technical results.

**Justification For Why Not Lower Score:**

N/A

---

### Decision · Program_Chairs · 2024-01-16

Reject